# Nonparametric Contextual Bandits
# in an Unknown Metric Space

**Nirandika Wanigasekara**
Computer Science
National University of Singapore
`nirandiw@comp.nus.edu.sg`

**Christina Lee Yu**
Operations Research and Information Engineering
Cornell University
`cleeyu@cornell.edu`

## Abstract

Consider a nonparametric contextual multi-arm bandit problem where each arm $a \in [K]$ is associated to a nonparametric reward function $f_a : [0,1] \to \mathbb{R}$ mapping from contexts to the expected reward. Suppose that there is a large set of arms, yet there is a simple but unknown structure amongst the arm reward functions, e.g. finite types or smooth with respect to an unknown metric space. We present a novel algorithm which learns data-driven similarities amongst the arms, in order to implement adaptive partitioning of the context-arm space for more efficient learning. We provide regret bounds along with simulations that highlight the algorithm's dependence on the local geometry of the reward functions.

## 1 Introduction

Contextual multi-arm bandits have been used to model the task of sequential decision making in which the rewards of different decisions must be learned over trial via trial-and-error. The decision maker receives reward for each of the arms (i.e. actions or options) she chooses across the time horizon $T$. In each trial $t$, the decision maker observes the context $x_t$, which represents the set of observable factors of the environment that could impact the performance of the action she chooses. The decision maker must select an action based on the context and all past observations. Upon choosing action $a \in [K]$, she observes a reward, which is assumed to be a stochastic observation of $f_a(x)$, the expected reward of action $a$ at context $x$. In each trial, she faces the dilemma of whether to choose an action in order to learn about its performance (i.e. exploration), or to choose an action that she believes will perform well as estimated from the limited previous data (i.e. exploitation).

Consider a setting when the number of actions is very large, e.g. there is a large number of users and products on an e-commerce platform such that fully exploring the entire space of possible recommendations is costly. It is often the case that there is additional structure amongst the large space of actions, which the algorithm could exploit to learn more efficiently. In real-world applications however, this additional structure is often unknown a priori and must be learned from the data, which itself could be costly as well. It becomes important to understand the tradeoff and costs of learning relationships amongst the arm from data over the course of the contextual bandit time horizon. We consider a stochastic nonparametric contextual bandit setting in which the algorithm is not given any information a priori about the relationship between the actions. The key question is: *Can an algorithm exploit hidden structure in a nonparametric contextual bandit problem with no a priori knowledge of the underlying metric?*

**Contributions** To our knowledge, we propose the first nonparametric contextual multi-arm bandit algorithm that incorporates latent arm similarities in a setting where no a priori information about the features or metric amongst the arms is given to the algorithm. The algorithm can learn more efficiently by sharing data across similar arms, but the tradeoff between the cost of estimating arm

similarities must be carefully accounted for. Our algorithm builds upon Slivkin's Zooming algorithm [22], adaptively partitioning the context-arm space using pairwise arm similarities estimated from the data. The adaptive partitioning allows the algorithm to naturally adapt the precision of its estimates around regions of the context-arm space that are nearly optimal, enabling the algorithm to more efficiently allocate its observations to regions of high reward.

We provide upper bounds on the regret that show the algorithm's dependence on the local geometry of the reward functions. If we let $f^*(x) := \max_{a \in [K]} f_a(x)$ denote the optimal reward at context $x$, then the regret depends on how the mass of the set $\{(a, x) : f^*(x) - f_a(x) \in (0, \delta]\}$ scales as $\delta$ goes to zero. This set represents the $\delta$-optimal region of the context-arm space except for the exactly optimal arms, i.e. the local measure of nearly optimal options centered around the optimal policy. The scaling of this set captures the notion of "gap" used in classical multi-arm bandit problems, but in the general contextual bandit setting with a large number of arms, it may be reasonable that the second optimal arm is very close in value to the optimal arm such that the gap is always very small. Instead the relevant quantity is the relative measure of arms that are $\delta$-optimal yet not optimal, i.e. have gap less than $\delta$. If the mass of such arms decreases linearly with respect to $\delta$, then we show that our algorithm achieves regret of $O(\sqrt{KT})$.

An interesting property of our algorithm is that it is fully data-dependent and thus does not depend on the chosen representation of the arm space. The arm similarities (or distances) are measured from data collected by the algorithm itself, and thus approximates a notion of distance that is defined with respect to the reward functions $\{f_a\}_{a \in [K]}$. The algorithm would perform the same for any permutation of the arms. In contrast, consider existing algorithms which assume a given distance metric or kernel function which the reward function is assumed to be smooth with respect to. Those algorithms are sensitive to the metric or kernel given to it, which itself could be expensive to learn or approximate from data. Suppose that nature applied a measure preserving transformation to the arm metric space such that the function is still Lipschitz but has a significantly larger Lispchitz constant. For example, consider a periodic function that repeats across the arm metric space. The performance of existing algorithms would degrade with poorer arm feature representations, whereas the algorithm we propose would remain agnostic to such manipulations.

We provide simulations that compare our algorithm to oracle variants that have special knowledge of the arms and a naive benchmark that learns over each arm separately. Initially our algorithm has a high cost due to learning the similarities, but for settings with a large number of arms and a long time horizon, the learned similarities pay off and improve the algorithm's long run performance.

**Related Work**  As there is a vast literature on multi-arm bandits, we specifically focus on literature related to the stochastic contextual bandit problem, with an emphasis on nonparametric models. In contextual bandits, in each trial the learner first observers a feature vector, refer to as "context", associated with each arm. The optimal reward is measured with respect to the context revealed at the beginning of each trial. One approach is to directly optimize and learn over a given space of policies rather than learn the reward functions [3, 5, 12, 14]. These methods do not require strict assumptions on the reward functions but instead depend on the complexity or size of the model class.

We focus on the alternative approach of approximating reward functions, which then depend on assumptions about the structure of the reward function. A common assumption to make is that the reward function is linear with respect to the observed context vector [15, 1, 2, 13], such that the reward estimation task reduces to learning coefficient vectors of the reward functions. [2] incorporates sparsity assumptions for the high dimensional covariate setting, and [13] imposes low rank assumptions on the coefficient vectors to reduce the effective dimension.

In the linear bandit setting with $K$ arms but only $\Theta$ arm types for $\Theta \ll K$, Gentile et al proposed an adaptive clustering algorithm which maintains an undirected graph between the arms and progressively erase edges when the estimated coefficient vectors of the pair of arms is above a set threshold [6]. Two arms of the same type are assumed to have the same coefficient vector. The threshold is chosen as a function of the minimum separation condition between coefficients vectors of different types, such that eventually the graph converges to $\Theta$ connected components corresponding to the $\Theta$ types. Collaborative filtering bandits [16] applies the same adaptive clustering concept to the recommendation system setting where both users and item types must be learned.

In the nonparametric setting, instead of fixing a parametric model class such as linear, most work imposes smoothness conditions on the reward functions, and subsequently use nonparametric estima-

tors such as histogram binning, $k$ nearest neighbor, or kernel methods to learn the reward functions [24, 20, 18, 19, 7]. As the contexts are observed, the estimator is applied to learn the reward of each arm separately, essentially assuming the number of arms is not too large. [7] provides an upper bound on regret of $\tilde{O}(KT^{\frac{d+1}{d+2}})$, where $d$ is the dimension of the context space, and $K$ is the number of arms.

The setting of continnum arm bandits has been introduced to approximate setting with very large action spaces. As there are infinitely many arms, it is common to impose smoothness with respect to some metric amongst the arms [17, 22, 8, 10, 9]. As the joint context-arm metric is known, these methods apply various smoothing techniques implemented via averaging datapoints with respect to a partitioning of the context-arm space, refining the smoothing parameter as more data is collected. [7] uses a $k$ nearest neighbor estimator using the joint context-arm metric. The contextual zooming algorithm from Slivkins [22] was a key inspiration for our proposed algorithm; it uses the given context-arm metric to *adaptively* partition the context-arm product space [22]. This enables the algorithm to efficiently allocate more observations to regions of the context-arm space that are near optimal. When $T$ is the time horizon and $d$ is the covering dimension of the context-arm product space, the regret of the contextual zooming algorithm is bounded above by $\tilde{O}(T^{\frac{d+1}{d+2}})$.

For settings with large but finite number of arms, there are nonparametric models which assume different types information is known about the relationship amongst the arms. Gaussian process bandits use a known covariance matrix to fit a Gaussian process over the joint context-arm space [11]. Taxonomy MAB assumes that similarity structure amongst the arm is given in terms of a hierarchical tree rather than metric [21]. Deshmukh et al assume that the kernel matrix between pairs of arms is known, and they subsequently use kernel methods to estimate the reward functions. Cesa-Bianchi et al assumes that a graph reflecting arm similarities is given to the algorithm, and their algorithm subsequently uses the Laplacian matrix of the given graph to regularize their estimates of the reward functions [4]. Wu et al assumes an influence matrix amongst arms is known and used to share datapoints among connected arms in the estimation [23]. A limitation of these approaches is that they assumes similarity information is provided to the algorithm either as a metric, kernel, or via a graph structure. In real-world applications, this similarity information is often not readily available and must be itself learned from the data.

## 2   Problem Statement

Assume that the context at each trial $t \in [T]$ is sampled independently and uniformly over the unit interval, $x_t \sim U(0,1)$, and revealed to the algorithm. Assume there are $K$ arms, or options, that the algorithm can choose amongst at each trial $t$. If the algorithm chooses arm $a_t$ at trial $t$, it observes and receives a reward $\pi_t \in \mathbb{R}$ according to $\pi_t = f_{a_t}(x_t) + \epsilon_t$, where $\epsilon_t \sim N(0, \sigma^2)$ is an i.i.d Gaussian noise term with mean 0 and variance $\sigma^2$, and $f_a(x)$ denotes the expected reward for arm $a$ as a function of the context $x$. We assume that each arm reward function $f_a : [0,1] \to [0,1]$ is $L$-Lipschitz, i.e. for all $x, x' \in [0,1]^2$, $|f_a(x) - f_a(x')| \leq L|x - x'|$.

The goal of our problem setting is to maximize the total expected payoff $\sum_{t=1}^{T} \pi_t$ over the time horizon $T$. We provide upper bounds on the expected contextual regret,

$$\mathbb{E}[R(T)] := \mathbb{E}\left[\sum_{t=1}^{T}(f^*(x_t) - f_{a_t}(x_t))\right] \quad \text{where} \quad f^*(x) := \max_{a \in [K]} f_a(x). \qquad (1)$$

We would like to understand whether an algorithm can efficiently exploit latent structure amongst the arm reward functions if it exists. Although the number of arms may be large, they could be drawn from a smaller set of finite arm types. Alternatively the arms could be draw from a continuum metric space such that the reward function is jointly Lipschitz over the context-arm space; however our algorithm would not have access to or knowledge of the underlying representation in the metric space.

## 3   Algorithm Intuition

We begin by describing an oracle algorithm that is given special knowledge of the relationship between the arms in the form of the context-arm metric. Assume that the arms are embedded into a metric space, and the function is Lipschitz with respect to that metric. The contextual zooming algorithm proposed by Slivkins in [22] reduces the large continuum arm set to the effective dimension

of the underlying metric space. Essentially, their model assumes that each arm is associated to some known parameter $\theta_a \in [0, 1]$, and that the expected joint payoff function is 1-Lipschitz continuous in the context-arm product space with respect to a known metric $\mathcal{D}$, such that for all context-arm pairs $(x, a)$ and $(x', a')$, $|f(x, \theta_a) - f(x', \theta_{a'})| \leq \mathcal{D}((x, \theta_a), (x', \theta_{a'}))$.

The key idea of Slivkin's zooming algorithm is to use adaptive discretization to encourage the algorithm to obtain more refined estimates in the nearly optimal regions of the space while allowing coarse estimates in suboptimal regions of the context-arm space. The algorithm maintains a partition of the context-arm space consisting of "balls", or sets, of various sizes. The algorithm estimates the reward function within a ball by averaging observed samples that lie within this ball. An upper confidence bound is obtained by accounting for the bias (proportional to the "diameter" of the ball due to Lipschitzness) and the variance due to averaging noisy observations within the ball. When a context arrives, the UCB rule is used to select a ball in the partition, and subsequently an arm in that ball. When the number of observations in a ball increases beyond the threshold such that the variance of the estimate is less than the bias, then the algorithm splits the ball into smaller balls, refining the partition locally in this region of the context-arm space.

The main intuition of the analysis is that the UCB selection rule guarantees (with high probability) that when a ball with diameter $\Delta$ is selected, the regret incurred by selecting this ball is bounded above by order $\Delta$. As a result, this algorithm is able to exploit the arm similarities via the joint metric in order to aggregate samples of similar arms such that the estimates will converge more quickly. Subsequently the algorithm refines the estimates and subpartitions the space as needed for regions that are near optimal and thus require tighter estimates in order to allow the algorithm to narrow in on the optimal arm. The limitation of the previous Zooming algorithm is that it depends crucially on the given knowledge of the context-arm joint metric, which could be unknown in advance.

**Arm Similarity Estimation**    In our model, we are not given any metric or features of the arm, thus the key question is whether it is still possible for an algorithm to exploit good structure amongst the arms if it exists. We propose an algorithm inspired by Slivkin's contextual zooming algorithm, which also adaptively partitions the context-arm space with the goal to allow for coarse estimates that converge quickly initially, and subsequently selectively refine the partition and the corresponding estimates in regions of the context-arm space that are nearly optimal. The key challenge to deal with is determining how to subpartition amongst the arms when we do not know any underlying metric or feature space. Our algorithm estimates a similarity (or distance) from the collected data itself, and uses the data-dependent distances to cluster/subpartition amongst the arms. This concept is similar to clustering bandits which also learns data-driven similarities, except that the clustering bandits works assume linear reward functions and finite types, whereas our model and algorithm is more general for nonparametric functions and arms drawn from an underlying continuous space [6].

We want our algorithm to partition the context-arm product space into balls, or subsets, within which the maximum diameter is bounded, where diameter of a subset is defined as $diam(\mathcal{S}) := \sup_{(x,a)\in\mathcal{S}} f_a(x) - \inf_{(x',a')\in\mathcal{S}} f_{a'}(x')$. We consider balls $\rho \subseteq [0, 1] \times [K]$ which have the form of $[c_0(\rho), c_1(\rho)] \times \mathcal{A}(\rho)$, where $c_0(\rho) \in [0, 1]$ denotes the start of the context interval, $c_1(\rho) \in [c_0(\rho), 1]$ denotes the end of the context interval, and $\mathcal{A}(\rho) \subseteq [K]$ denotes the subset of arms. We use $\Delta(\rho) := c_1(\rho) - c_0(\rho)$ to denote the "width" of the context interval pertaining to the ball $\rho$.

In order to figure out which set of arms to include in a "ball" such that the diameter is bounded, we ideally would like to measure the $L_\infty$ distance with respect to the context interval of the ball, $\max_{x\in[c_0(\rho),c_1(\rho)]} |f_a(x) - f_{a'}(x)|$. As the functions are assumed to be Lipschitz with respect to the context space, a bound on the $L_2$ distance also implies a bound on the $L_\infty$. Our algorithm approximates the $L_2$ distance, defined with respect to an interval $[u, v]$ according to

$$\mathcal{D}_u^v(a, a') := \sqrt{\tfrac{1}{200} \sum_{i\in[200]} \left( f_a(z_i(u, v)) - f_{a'}(z_i(u, v)) \right)^2} \tag{2}$$

where $z_i(u, v) = \left(1 - \tfrac{i}{200}\right) u + \tfrac{i}{200} v$. This is a finite sum approximation to the integrated $L_2$ distance between $f_a$ and $f_{a'}$ within the interval $[u, v]$.

Our algorithm uses the data collected for an arm in order to approximate the reward functions using a $k$ nearest neighbor estimator, and subsequently uses the estimated reward functions to approximate $\mathcal{D}_u^v$. These approximate distances are then used to cluster the arms when subpartitioning. With high probability, we show that the diameter of the constructed balls is bounded by $2L\Delta(\rho)$. Our algorithm

collects extra samples to compute these distances, and a key part of the analysis is to understand when the improvement in the learning rate of the reward functions is sufficient enough to offset the cost of estimating arm distances.

## 4    Algorithm Statement

Let $n_t(\rho) = \sum_{s=1}^{t-1} \mathbb{I}(\rho_s = \rho)$ denote the number of times $\rho$ has been selected before trial $t$. Let $\mu_t(\rho) = \frac{1}{n_t(\rho)} \sum_{s=1}^{t-1} \mathbb{I}(\rho_s = \rho) \pi_s$ denote the average observed reward from $\rho$ before trial $t$. Define

$$UCB_t(\rho) = \mu_t(\rho) + 2L\Delta(\rho) + \sqrt{6\sigma^2 \ln(T)/n_t(\rho)}, \tag{3}$$

which gives an upper confidence bound for the maximum reward achievable by any context-arm pair in the ball $\rho$. The algorithm maintains two sets of balls, $\mathcal{P}$ and $\mathcal{P}^*$, such that $\mathcal{P} \cup \mathcal{P}^*$ is a partition of the context-arm space, i.e. all balls are disjoint and the union cover the entire space. We refer to balls in $\mathcal{P}^*$ as flagged. They are given ultimate priority in the algorithm, until sufficient samples are collected to further subpartition this ball via clustering. We refer to balls in $\mathcal{P}$ as "active", within which priority is given to balls with higher upper confidence bound (UCB).

**Ball-Arm Selection Rule**    In a given trial $t$, when the context $x_t$ arrives, the algorithm identifies the flagged balls $\rho \in \mathcal{P}^*$ which contain context $x_t$, i.e. $x_t \in [c_0(\rho), c_1(\rho)]$, and gives priority amongst them to balls with larger width $\Delta(\rho)$,

$$\rho_t = \operatorname{argmax}_{\rho \in \mathcal{P}^*} \Delta(\rho) \mathbb{I}(x_t \in [c_0(\rho), c_1(\rho)]).$$

If there are no flagged balls in $\mathcal{P}^*$ which contain $x_t$, then the algorithm selects an active ball $\rho \in \mathcal{P}$ containing $x_t$, and gives priority to the ball with the highest upper confidence bound $UCB_t(\rho)$,

$$\rho_t = \operatorname{argmax}_{\rho \in \mathcal{P}} UCB_t(\rho) \mathbb{I}(x_t \in [c_0(\rho), c_1(\rho)]). \tag{4}$$

When a ball $\rho_t$ is chosen, the algorithm plays an arm $a_t \in \mathcal{A}_{\rho_t}$ via a round robin ordering. The algorithm observes a noisy reward $\pi_t$ for arm $a_t$ and updates $n_t(\rho)$, $\mu_t(\rho)$, and $UCB_t(\rho)$ accordingly.

By grouping the context-arm pairs into balls, the algorithm aggregates the observed rewards within a ball to trade-off between bias and variance. For any given trial, the algorithm reduces the decision problem from selecting amongst a large number of arms to selecting amongst a smaller set of balls, which each consist of a subset of arms. Whenever the ball is subpartitioned, the width of the context interval is halved, such that balls never repeat, and are always strictly nested within a hierarchy. Furthermore, the fact that the algorithm gives priority to flagged balls with larger context widths implies that the data collected in the "flagged" phase of every ball will be uniformly distributed over context width of that ball.

**Flagging Rule**    At the beginning of the algorithm, the entire context-arm space is flagged as a single large ball to be subpartitioned, i.e. $\mathcal{P}^* = \{([0,1] \times [K])\}$ and $\mathcal{P} = \emptyset$. In subsequent rounds, we flag a ball $\rho \in \mathcal{P}$ whenever it satisfies the condition $n_t(\rho_t) > 6\sigma^2 \ln(T)/L^2 \Delta^2(\rho_t)$. Upon being flagged, $\rho$ is removed from $\mathcal{P}$ and added to $\mathcal{P}^*$. Let stopping time $\tau_f(\rho)$ denote the trial $t$ that ball $\rho$ is flagged. Intuitively, the threshold is chosen at a point where the confidence radius, i.e. natural variation in the estimates due to the additive Gaussian observation error, is on the order of the diameter of the ball. As a result, further collecting samples does not improve the overall UCB because the diameter of the ball will dominate the expression.

**Sub-Partitioning via Clustering**    Recall that flagged balls in $\mathcal{P}^*$ are always given priority over active balls in $\mathcal{P}$. The observations collected in the flagged phase are used to estimate distances, or similarities between the arms for the purpose of subpartitioning the ball into smaller balls. In particular, the algorithm splits the context space $[c_0(\rho), c_1(\rho)]$ into 64 evenly sized intervals and waits until it collects at least $k$ samples within each of the 64 intervals for each of the arms $a \in \mathcal{A}(\rho)$, where $k$ is chosen according to $k = 5431\sigma^2 \ln(T|\mathcal{A}(\rho)|)/(L^2 \Delta^2(\rho))$. This condition is mathematically stated as $\prod_{a \in \mathcal{A}(\rho)} \text{SUFFDATA}(a) == 1$ where

$$\text{SUFFDATA}(a) := \prod_{i=1}^{64} \mathbb{I}\left(\sum_{s > \tau_f(\rho)} \mathbb{I}(\rho_s = \rho, a_s = a) \mathbb{I}(x_s \in [w_{i-1}, w_i]) \geq k\right),$$

for $w_i = c_0(\rho) + i\Delta(\rho)/64$. When this sufficient data condition is satisfied, the algorithm uses the observations collected in the flagged phase to compute pairwise arm distances approximating (2). Let $\tau_{cl}(\rho)$ denote the trial in which the sufficient data condition is satisfied and $\rho$ is subpartitioned.

The SUBPARTITION subroutine estimates the reward functions via a $k$-nearest neighbor estimator,

$$\hat{f}_a(x) = \frac{1}{k} \sum_{s=\tau_f(\rho)+1}^{\tau_{cl}(\rho)} \mathbb{I}\left(\rho_s = \rho, a_s = a\right) \mathbb{I}\left(x_s \in \text{ k-NN}\right) \pi_s, \qquad (5)$$

where $x_s$ is a $k$ nearest neighbor of $x$ if $\sum_{\ell=\tau_f(\rho)+1}^{\tau_{cl}(\rho)} \mathbb{I}\left(\rho_\ell = \rho, a_\ell = a\right) \mathbb{I}\left(|x_\ell - x| \le |x_s - x|\right) \le k$.

Given the estimated functions $\{\hat{f}_a\}_{a \in \mathcal{A}(\rho)}$ and a pair of arms $a, a' \in \mathcal{A}(\rho)$, we compute $\hat{\mathcal{D}}_u^v(a, a')$ for intervals $[u, v] = [c_0(\rho), (c_0(\rho) + c_1(\rho))/2]$ and $[u, v] = [(c_0(\rho) + c_1(\rho))/2, c_1(\rho)]$ according to

$$\hat{\mathcal{D}}_u^v(a, a') := \sqrt{\frac{1}{200} \sum_{i \in [200]} \left(\hat{f}_a(z_i(u, v)) - \hat{f}_{a'}(z_i(u, v))\right)^2 - \frac{2\sigma^2}{k}} \qquad (6)$$

where $z_i(u, v) = \left(1 - \frac{i}{200}\right) u + \frac{i}{200} v$ and the term $\frac{2\sigma^2}{k}$ accounts for bias due to the noise.

We use the computed distances $\hat{\mathcal{D}}_u^v(a, a')$ to subpartition $\rho$ by clustering the arms for each half of the context interval separately. For an arbitray ordering of the arms, we test if the next arm has distance less than $3L(v - u)/16$ to any of the existing cluster centers. If so, we assign it to the cluster associated to the closest cluster center. Otherwise, we create a new cluster and assign this arm to be the cluster center. This results in a clustering in which all pairs of cluster centers are guaranteed to be distance $3L(v - u)/16$ apart, and all members of a cluster must be within distance $3L(v - u)/16$ to the cluster center. These distances are measured with respect to the data dependent estimates $\hat{\mathcal{D}}_u^v(a, a')$. In our analysis, we show that with high probability $\hat{\mathcal{D}}_u^v(a, a') \approx \mathcal{D}_u^v(a, a')$.

Once the clusters are created, then $\rho$ is unflagged (removed from $\mathcal{P}^*$) and new balls corresponding to each of the clusters for each half of the context interval are added to the active set $\mathcal{P}$. See the appendix for a pseudocode description of the algorithm.

## 5  Simulation

We test our algorithm on a model with 50, 100, 200 arms and a context space of $[0, 1]$. Each arm $a$ corresponds to a parameter $\theta_a$ uniformly spaced out within $[0, 1]$. The expected reward for arm $a$ and context $x$ is

$$f_a(x) := g(x, \theta_a) = 1 - \left|x - 4 \min_{z \in \{0, 0.5, 1\}} |\theta_a - z|\right|.$$

This function is periodic with respect to $\theta$, and can be depicted as a zigzag. Our distance estimate $\hat{\mathcal{D}}_u^v(a, a')$ approximates $\mathcal{D}_u^v(a, a')$, which is defined with respect $f_a$ and $f_{a'}$ directly and does not depend on $\theta_a$. Consider a measure preserving transformation that maps $\theta_a$ to $\phi_a = 4 \min_{z \in \{0, 0.5, 1\}} |\theta_a - z|$, such that the reward function is equivalently described by $f_a(x) = 1 - |x - \phi_a|$. An algorithm which partitions with respect to $\mathcal{D}_u^v(a, a')$ would be agnostic to such a transformation, as opposed to an algorithm which depends on a metric defined with respect the arm's representation, which would perform worse on $\theta_a$ than $\phi_a$.

We benchmark the performance of our Approx-Zooming algorithm against three variations:

- *Approx-Zooming -With-True-Reward-Function*: We give the Approx-Zooming algorithm oracle access to evaluate $\mathcal{D}_u^v(a, a')$ at no cost, which is used to subpartition whenever a ball is flagged.
- *Approx-Zooming -With-Similarity-Metric*: We give the Approx-Zooming algorithm oracle access to evaluate $|\theta_a - \theta_{a'}|$ at no cost, which is used to subpartition whenever a ball is flagged.
- *Approx-Zooming -With-No-Arm-Similarity*: This naive variant uses no arm similarities, estimating each arm's reward independently. The context space is adaptively partitioned via our algorithm.

We chose the model parameters that led to the highest average cumulative reward in each baseline algorithm. For all algorithms the flagging rule is set to $n_t(\rho) \ge 4 \ln(T)/\Delta^2$, and $\sigma$ was set to $1e - 2$. For Approx-Zooming , $k$ was set to $10$. We set the number of trials $T$ to $100,000$ as all the algorithms had converged to their optimal point by then. Additional details on how the model parameters were chosen is given in Appendix F.

In figure 1, we plot the average cumulative reward over the trials, i.e. $\frac{1}{T}\sum_{t=1}^{T}\pi_t$, where $T$ is the total number of trials and $\pi_t \in (0,1)$ is the reward observed in the $t^{\text{th}}$ trial. We plot the result for the 200 arm setting with $\sigma$ set to $1e - 2$. As we can see, the oracle variant of the algorithm that uses the true reward function to calculate $\mathcal{D}_u^v(a, a')$ achieves the best cummulative reward across the entire time horizon. Not surprisingly, the algorithm which learns each arm separately takes more time to converge to the optimal policy compared to all the other methods. Our Approx-Zooming algorithm has a heavy cost up front due to the clustering of the arms globally, but the algorithm improves over the time horizon as it learns the correct arm similarities. The oracle variant which uses the similarity metric $|\theta_a - \theta_{a'}|$ performs worse than the true $\mathcal{D}_u^v(a, a')$ variant, as it does not account for the periodic nature of the function. This supports our intuition that algorithms which depend on a given metric are sensitive to the choice of a good vs bad metric.

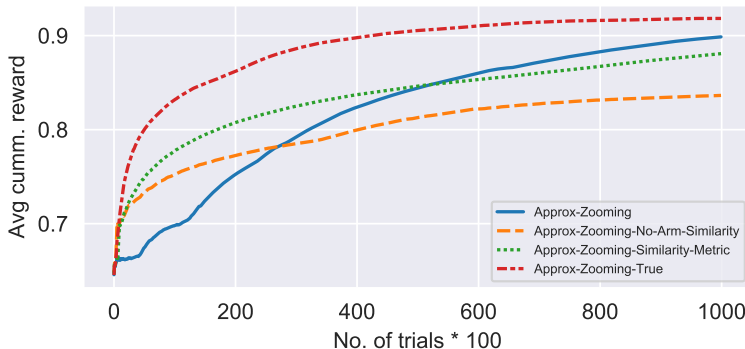

Figure 1: Avg. cumulative reward vs. number of trials

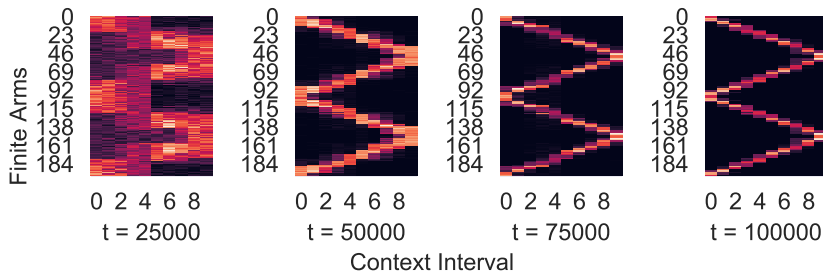

Figure 2: Approx-Zooming Selected Arm Frequency Over The Trials

In figure 2 we plot the frequencies an arm is selected in different contexts over the $T$ trials. Each of the four plots corresponds to averaging the frequency over $T/4$ trials across the time horizon. The x-axis refers to the context space, and the y-axis refers to the set of arms. Initially the frequency plot is very blurry, indicating that our algorithm is not necessarily playing the optimal arm but selecting arms to learn the latent arm structure. As time progresses our algorithm learns the similarities amongst arms and gradually plays the arms using the latent structure, which is depicted by the zigzag shape sharpening. Finally, in the last trials Approx-Zooming plays the optimal policy, which corresponds to the clear zigzag. In Appendix F we present similar plots for the benchmark algorithms.

Our simulations show that when the number of arms is large, it is important to use similarities amongst arms to more quickly learn the optimal policy. In addition our results highlight the fact that metric-based algorithms may be sensitive to the choice of metric, which is not a trivial task. In contrast, our approach relies on samples from the reward distribution to learn the latent structure, and is thus agnostic to any choice of metric. However, the parameter $k$ needs to be carefully tuned for our algorithm to avoid unnecessary sampling for estimating similarities. In Appendix F we include similar plots for other parameters of the problem, in particular for smaller number of arms. We see that for 50 arms or 100 arms, the cost due to the added extra exploration may exceed the gain from learning the metric, and thus we anticipate that the benefits of learning the metric only dominates in regimes where the number of arms is large and the time horizon is sufficiently long.

# 6 Upper bound on the Regret

We present a general bound on the regret expressed as a function of a quantity relating to the local geometry of the reward function nearby the optimal policy. Let us denote $w_i(\ell) = [(\ell-1)2^{-i}, \ell 2^{-i}]$, $\kappa(x) = f^*(x) - \max_{a \in [K]} f_a(x) \, \mathbb{I}(f_a(x) \neq f^*(x))$, and

$$M_i = \sum_{\ell=1}^{2^i} \mathbb{I}\left(\min_{x \in w_i(\ell)} \kappa(x) \leq 20 \, L \, 2^{-i}\right) \sum_{a \in [K]} \mathbb{I}\left((f^*(2^{-i}\ell) - f_a(2^{-i}\ell)) \leq 22 \, L \, 2^{-i}\right).$$

**Theorem 6.1.** *The expected contextual regret of Approx-Zooming is bounded above by*

$$\mathbb{E}\left[R(T)\right] = O\left(\sigma^2 L^{-2} K \ln(TK) + \min_{i_{\max} \in \mathbb{Z}_+} \left(LT2^{-i_{\max}} + \sum_{i=1}^{i_{max}-1} \sigma^2 L^{-1} M_i 2^i \ln(TK)\right)\right).$$

The analysis relies on showing that the instantaneous regret incurred by choosing a ball with context width $\Delta$ is bounded above by $O(L\Delta)$. The first term in the regret is due to the very first intitial clustering phase. The second term $L \, T \, 2^{-i_{\max}}$ bounds the regret incurred by all balls with context width at most $2^{-i_{\max}}$. The terms in the summation bound the regret incurred by balls with context width equal to $2^{-i}$. The function $\kappa(x)$ represents the lowest regret achieved by the second-most optimal arm, which lower bounds the suboptimality gap. In alignment with our intuition from classical MAB, when the suboptimality gap is large, the algorithm is able to more quickly converge to the optimal arm at context $x$. When we bound the regret incurred by all balls with context width $2^{-i}$, we can thus remove subintervals of the context for which $\kappa(x)$ is large as the algorithm will have already converged to the optimal arm. This is reflected in the first indicator function within the expression $M_i$. Once restricted to context subintervals where the suboptimality gap is not too large, the expression $\sum_{a \in [K]} \mathbb{I}\left((f^*(2^{-i}\ell) - f_a(2^{-i}\ell)) \leq 22 \, L \, 2^{-i}\right)$ counts the number of arms for which the suboptimality gap is at most $22 \, L \, 2^{-i}$; arms for which the suboptimality gap is larger will have already been deemed suboptimal. As the specific bounds on $M_i$ depend on the model and local geometry amongst the arms, we provide bounds for two concrete examples to give more intuition.

**Finite Types** Suppose that the reward functions for the $K$ arms, $\{f_a\}_{a \in [K]}$ only takes $\Theta$ different values. Essentially, this implies that there are $\Theta$ different types of arms, but we don't know the arm types a priori. Within each type, the reward function is exactly the same. Let us define

$$\mu_\kappa(z) := \mu(\{x \in [0,1] \; s.t. \; \kappa(x) \leq z\})$$

where $\mu$ is the Lebesgue measure. Then we can show that $M_i \leq 2^i K \mu_\kappa(22 \, L \, 2^{-i})$. The regret is bounded by the local measure function $\mu_\kappa$. In the finite types setting, the optimal policy corresponds to partitioning the context space $[0,1]$ into a set of intervals, $\mathcal{S}^*$, such that across each interval $\int \in \mathcal{S}^*$, the optimal policy does not change. Let us consider the setting that $\kappa(x)$ decreases linearly fast nearby the points where the optimal policy changes, so that for some constant $L'$, $\mu_\kappa(22 \, L \, 2^{-i}) \leq 22 \, |\mathcal{S}^*| \, L \, 2^{-i}/L'$. By plugging the bound on $M_i$ into the main theorem and choosing $i_{\max} = \log(L'LT/22\sigma^2|\mathcal{S}^*|K \ln(TK))/2$, it follows that

$$\mathbb{E}\left[R(T)\right] \leq O\left(\sigma^2 L^{-2} K \ln(TK) + \sqrt{\sigma^2 |\mathcal{S}^*| LL'^{-1} TK \ln(TK)}\right) \tag{7}$$

**Lipschitz with respect to continuous arm metric space** Suppose that each arm $a$ is associated to a latent feature $\theta_a \in [0,1]$, and the expected reward function $f_a(x) = g(x, \theta_a)$, where $g : [0,1] \times [0,1] \to [0,1]$ is a $L$-Lipschitz function with respect to both the contexts and the arm latent features such that $|g(x, \theta) - g(x', \theta')| \leq L(|x - x'| + |\theta - \theta'|)$. If we assume that the arm latent features are uniformly spread out, $\{\theta_a\} = \{i/K\}_{i \in [K]}$, then

$$M_i \leq \sum_{j \in [K]} \sum_{\ell \in [2^i]} \mathbb{I}\left((f^*(2^{-i}\ell) - g(2^{-i}\ell, \tfrac{j}{K})) \leq 22 \, L \, 2^{-i}\right), \tag{8}$$

which is a discrete approximation to the area of the context-arm space for which the suboptimality gap is at most $22 \, L \, 2^{-i}$. We can visualize $\sum_{\ell=1}^{2^i} M_i(\ell)$ by considering the contour plot of $f^*(x) - g(x, \theta)$, and counting how many grid points $\{(2^{-i}\ell, j/K)\}_{\ell \in [2^i], j \in [K]}$ are lower than $22L2^{-i}$. For large $i$ and $K$, this is approximately $2^i K \mu(\{(x, \theta) : g(x, \theta) - f^*(x) \geq -22L2^{-i}\})$, where $\mu$ is the Lebesgue measure. The curve at the lowest level of the contour plot corresponds to the set $\{(x, \theta) \; s.t. \; g(x, \theta) - $

$f^*(x) = 0$}, which contains for each context the set of arm features that optimize the expected reward. The final regret depends on the local measure of the joint reward function.

As an example, if we consider the reward function $g(x, \theta) = 1 - L|x - \theta|$ for some $L \in (0, 1)$, we can show that $M_i \leq 44K$, i.e. it is bounded by a constant with respect to $i$. Therefore by plugging into the main theorem and choosing $i_{\max} = \log\left(20L^2T/\sigma^2K\ln(TK)\right)/2$ it follows that

$$\mathbb{E}\left[R(T)\right] \leq O\left(\sigma^2L^{-2}K\ln(TK) + \sqrt{\sigma^2KT\ln(TK)}\right). \tag{9}$$

## 7 Discussion

**Interpreting the results.** We began this paper with the question: *Can an algorithm exploit hidden structure in a nonparametric contextual bandit problem with no a priori knowledge of the underlying metric?* The results of our simulations suggest that our proposed algorithm (with empirically tuned hyperparameters) can perform better than the corresponding algorithm that learns over each arm separately, or that uses a suboptimal metric. However, the regret bounds we present are not sufficiently strong to provably show that the algorithm outperforms learning on arms separately. The stated upper bound on regret in [7] is linear in the number of arms $K$, however this may be simply due to the fact that they did not optimize with respect to $K$ in their analysis. Our regret bound is most comparable to the regret for the infinite arm setting presented in [22], and it can be recovered from their bound by imposing the discrete metric amongst the arms.

**Generalizing to higher context dimension.** For simplicity, we have stated our algorithm and analysis for the 1D context space, but the results extend to the general $d$-dimensional setting. The only change required algorithmically is in the subpartitioning/clustering step. Let us define $C_d(q)$ to be the number of balls of radius $r/q$ needed to cover a ball of radius $r$, which scales exponentially in the dimension $d$, e.g. $q^d$. Since we are now estimating the reward function $f$ over a $d$-dimensional context space, the number of sub-regions of the context space that need to be clustered will be $C_d(2)$, and the number of samples needed to guarantee that the $k$-nearest neighbor samples are within distance $\frac{1}{16}$ radius, will be equal to $\tilde{O}(kC_d(32))$. To compute $\hat{\mathcal{D}}$, we will instead have a $d$-dimensional summation over the subset of the context space. Once $\hat{\mathcal{D}}$ is computed, then the clustering of arms will have the same computational cost, i.e. linear in number of arms to be clustered. The analysis can be modified to account for the $d$-dimensional setting, and the final regret bound will look like

$$O\left(C_d(2)C_d(32)\sigma^2L^{-2}K\ln(TK) + \min_{i_{\max}\in\mathbb{Z}_+}\left(LT2^{-i_{\max}} + \sum_{i=1}^{i_{max}-1}C_d(2)C_d(32)\sigma^2L^{-1}M_i2^i\ln(TK))\right)\right),$$

where $M_i$ instead sums over an $\epsilon$-net of the context space for $\epsilon = 2^{-i}$, and thus we may expect $M_i$ to grow exponentially in $i \times d$, depending on the distribution of the reward function and the finite arms. The growth of $M_i$ will dominate the regret bound with respect to the dependence on the dimension $d$.

**Choice of metric.** Nature could apply a measure preserving transformation to the arm metric space such that the joint function has a significantly higher Lipschitz constant. This representation would incur a worse performance by the previous Zooming algorithm, indicating that the algorithm is critically dependent on the choice of representation and metric. As an example, suppose that arms are each associated to some latent parameter $\theta \in (0, 2\pi)$, and the reward function associated to an arm $a$ is $f_a(x) = x + \sin(L\theta_a)$. The Lipschitz constant with respect to $\theta$ is $L$. By applying a change of variables from $\theta$ to $t(\theta) = L\theta \mod 2\pi$, the associated reward function in terms of the representation $t_a = t(\theta_a)$ would be $f_a(x) = x + \sin(t_a)$, which only has Lipschitz constant 1 with respect to the reparamerization $t$. This is only a simple example amongst many that illustrate the importance of the choice of metric for learning. In contrast, our algorithm estimates similarity amongst the arms directly from data collected from the reward functions, which essentially estimates distance in the function space; as a result our algorithm is invariant to any specific covariate representation.

**Future Work.** The current results are stated only for Lipschitz reward functions, where the tuning parameters depend on the Lipschitz constant. It may be interesting to generalize the algorithm to Holder continuous reward functions, and consider how to adapt the algorithm to the smoothness parameters if unknown. It would also be interesting to explore the connections to Gaussian process bandits. One would need to specify the covariance matrix amongst arms, and it may be possible to consider empirically estimating the covariance matrix in the process of learning.

## Acknowledgement

This work is supported by the National University of Singapore and A*STAR - SERC PSF Grant 1521200084. We thank Professor David S. Rosenblum for his support of this project through insightful discussions and feedback.

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
