[Supplementary Material]

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

## Footnotes

[1]The code to the algorithms and simulation is avialble at `https://bitbucket.org/nirandiw/context_similarity/src/master/`

[2]The hyper parameter tuning results are available at `https://drive.google.com/open?id=1vb-RK8E_flPZ7haN_M83Y63Fes87j6fs`

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

# A Algorithm Notation and Pseudocode

The algorithm is presented in full in the main body of the paper, but for reference we have also included pseudocode below.

---

**Algorithm 1** Approx-Zooming Algorithm

---

**Require:** context space $[0,1]$, arm space $[K]$, Lipschitz constant $L$ and time horizon $T$

1: **function** APPROX-ZOOMING (K, L, T)
2:      Parameter $k = \frac{5431\sigma^2 \ln(T|\mathcal{A}(\rho)|)}{L^2\Delta^2(\rho)}$
3:      Initialize $\mathcal{P}^* = \{([0,1] \times [K])\}, \mathcal{P} = \emptyset$
4:      **for** $t \in [T]$ **do**
5:         Context $x_t$ "arrives"
6:         **if** $x_t \in [c_0(\rho), c_1(\rho)]$ for any $\rho \in \mathcal{P}^*$ **then**          $\triangleright$ Check for any flagged balls
7:            $\rho_t = \operatorname{argmax}_{\rho \in \mathcal{P}^*} \Delta(\rho)\mathbb{I}(x_t \in [c_0(\rho, c_1(\rho)])$
8:            Select $a_t = \min\{a \in \mathcal{A}(\rho) \text{ s.t. } \text{SUFFDATA}(a) = 0\}$
9:            Play arm $a_t$ and observe payoff $\pi_t = f_{a_t}(x_t) + \epsilon_t$
10:           **if** $\prod_{a \in \mathcal{A}(\rho)} \text{SUFFDATA}(a) == 1,$ **then**
11:              new partitions = SUBPARTITION$(\rho)$
12:              $\mathcal{P} = \mathcal{P} \cup$ new partitions
13:              $\mathcal{P}^* = \mathcal{P}^* \setminus \rho$
14:         **else**          $\triangleright$ If no relevant flagged balls, then use UCB selection rule
15:            $\rho_t = \operatorname{argmax}_{\rho \in \mathcal{P}} UCB_t(\rho)\mathbb{I}(x_t \in [c_0(\rho), c_1(\rho)])$
16:            Play any arm $a_t \in \mathcal{A}(\rho_t)$ and observe payoff $\pi_t = f_{a_t}(x_t) + \epsilon_t$
17:           **if** $n_t(\rho_t) > \frac{6\sigma^2 \ln(T)}{L^2\Delta^2(\rho_t)}$ **then**
18:              $\mathcal{P} = \mathcal{P} \setminus \{\rho_t\}, \mathcal{P}^* = \mathcal{P}^* \cup \{\rho_t\}$ $\triangleright$ Remove from active set, flag for subpartition
19:              $\tau_f(\rho) = t$
20:
21: **function** SUBPARTITION$(\rho)$
22:      Initialize centers $\mathcal{C}_0 = \emptyset, \mathcal{C}_1 = \emptyset$
23:      **for** $\ell \in \{0,1\}$ **do**
24:         $u = c_0(\rho) + \frac{\Delta(\rho)\ell}{2}, v = c_0(\rho) + \frac{\Delta(\rho)(1+\ell)}{2}$
25:         **for** $a \in \mathcal{A}_\rho$ in arbitrary order **do**
26:            Compute $\hat{D}_u^v(a,y)$ for all $y \in \mathcal{C}_\ell$
27:            **if** $\min_{y \in \mathcal{C}_\ell} \hat{\mathcal{D}}_u^v(a,y) > \frac{3}{16}L(v-u)$ **then**
28:              Add $a$ as a new center, $\mathcal{C}_\ell = \mathcal{C}_\ell \cup \{a\}, \mathcal{S}_\ell(a) = \{a\}$
29:            **else**
30:              $y = \operatorname{argmin}_{y \in \mathcal{C}_\ell} \hat{D}_\rho(a,y)$
31:              $\mathcal{S}_\ell(y) = \mathcal{S}_\ell(y) \cup \{a\}$
32:      **return** $\left\{ \left([c_0(\rho), \frac{c_0(\rho)+c_1(\rho)}{2}] \times \mathcal{S}_0(y)\right) \right\}_{y \in \mathcal{C}_0} \cup \left\{ \left([\frac{c_0(\rho)+c_1(\rho)}{2}, c_1(\rho)] \times \mathcal{S}_1(y)\right) \right\}_{y \in \mathcal{C}_1}$

---

Recall the notation and definitions introduced:

- $diam(\mathcal{S}) := \sup_{(x,a) \in \mathcal{S}} f_a(x) - \inf_{(x',a') \in \mathcal{S}} f_{a'}(x')$
- $[c_0(\rho), c_1(\rho)]$ denotes the context interval of ball $\rho$.
- $\mathcal{A}(\rho)$ denotes the set of arms in ball $\rho$.
- $\Delta(\rho) := c_1(\rho) - c_0(\rho)$ denotes the context width of ball $\rho$.
- $n_t(\rho) := \sum_{s=1}^{t-1} \mathbb{I}(\rho_s = \rho)$ denotes the number of trials ball $\rho$ has been chosen by the algorithm before trial $t$.
- $\mu_t(\rho) := \frac{1}{n_t(\rho)} \sum_{s=1}^{t-1} \mathbb{I}(\rho_s = \rho) \pi_s$ denotes the average observed reward from this ball before trial $t$.
- $UCB_t(\rho) := \mu_t(\rho) + 2L\Delta(\rho) + \sqrt{\frac{6\sigma^2 \ln(T)}{n_t(\rho)}}$ is an upper confidence bound for the maximum reward achievable by any context-arm pair in the ball $\rho$.

- $\tau_f(\rho)$ denotes the trial that ball $\rho$ is flagged.
- $\tau_{cl}(\rho)$ denotes the trial in which the SUBPARTITION subroutine is called.
- $\text{SUFFDATA}(a) := \prod_{i=1}^{64} \mathbb{I}\left( \sum_{s > \tau_f(\rho)} \mathbb{I}(\rho_s = \rho, a_s = a) \, \mathbb{I}(x_s \in [w_{i-1}, w_i]) \geq k \right)$ for $w_i = c_0(\rho) + \frac{i\Delta(\rho)}{64}$ and $k = 5431\sigma^2 \ln(T|\mathcal{A}(\rho)|)/(L^2\Delta^2(\rho))$.
- The reward function for an arm $a$ is estimated via a $k$-NN estimator

$$\hat{f}_a(x) = \frac{1}{k} \sum_{s=\tau_f(\rho)+1}^{\tau_{cl}(\rho)} \mathbb{I}(\rho_s = \rho, a_s = a) \, \mathbb{I}(x_s \in \text{ k-NN}) \, \pi_s$$

  where $x_s$ is a $k$ nearest neighbor datapoint for computing $\hat{f}_a(x)$ if

$$\mathbb{I}(x_s \in \text{ k-NN}) = \sum_{\ell=\tau_f(\rho)+1}^{\tau_{cl}(\rho)} \mathbb{I}(\rho_\ell = \rho, a_\ell = a) \, \mathbb{I}(|x_\ell - x| \leq |x_s - x|) \leq k.$$

- The distance between arms $a$ and $a'$ for interval $[u, v]$ is estimated according to

$$\hat{\mathcal{D}}_u^v(a, a') := \sqrt{ \frac{1}{200} \sum_{i \in [200]} \left( \hat{f}_a(z_i(u, v)) - \hat{f}_{a'}(z_i(u, v)) \right)^2 - \frac{2\sigma^2}{k} }$$

  where $z_i(u, v) = \left(1 - \frac{i}{200}\right) r + \frac{i}{200} s$.

# B  Proof Sketch

Our algorithm and analysis take after the Zooming algorithm [9, 22]. However, the major difference is that their model assumes the metric is directly known in advance, but our algorithm must learn the metric. In particular, each trial we subpartition a set into finer clusters, we pay extra cost to collect samples to estimate distances used for determining the subsequent clusters.

**Lemma B.1.** *With probability at least $1 - 2T^{-1}$, over the entire course of the algorithm for all $t \in [T]$ and $\rho \in \mathcal{P}$,*

$$\mu_t(\rho) \in \left[ \min_{(x,a) \in \rho} f_a(x) - \sqrt{\frac{6\sigma^2 \ln(T)}{n_t(\rho)}}, \; \max_{(x,a) \in \rho} f_a(x) + \sqrt{\frac{6\sigma^2 \ln(T)}{n_t(\rho)}} \right].$$

**Lemma B.2.** *With probability at least $1 - 4T^{-1}$, over the entire course of the algorithm for all trials that $\hat{\mathcal{D}}_u^v(a, y)$ is evaluated,*

$$\left\{ |\hat{\mathcal{D}}_u^v(a, y) - \mathcal{D}_u^v(a, y)| \leq \frac{1}{8} L(v - u) \right\}.$$

We refer to the set of conditions in Lemmas B.1 and B.1 as the "good event" and denote them with $\mathcal{G}$.

Argue that the resulting clustering produced by SUBPARTITION satisfies that all pairs in the same cluster must be "close".

**Lemma B.3.** *Conditioned on the "good events" $\mathcal{G}$, for any $\rho \in \mathcal{P}$ at any point of the algorithm, $diam(\rho) \leq 2L\Delta(\rho)$.*

Let us define

$$gap(\rho) = \min_{(x,a) \in \rho} (f^*(x) - f_a(x)).$$

Conditioned on good events above, for any $\rho \in \mathcal{P}_t$, we upper bound the total number of trials this set can be chosen by the algorithm until either it is completely dominated, or it is flagged to be subpartitioned.

**Lemma B.4.** *Conditioned on the "good events" $\mathcal{G}$, if ball $\rho$ is chosen by the algorithm at trial $t$ via the UCB rule, then*

$$n_t(\rho) \leq \min\left\{ \frac{24\sigma^2 \ln(T)}{(gap(\rho) - 4L\Delta(\rho_t))^2}, \frac{6\sigma^2 \ln(T)}{L^2\Delta^2(\rho)} + 1 \right\} \tag{10}$$

*Therefore the max number of trials ball $\rho$ is chosen via the UCB rule is bounded above by,*

$$\sum_{t=1}^{\tau_f(\rho)} \mathbb{I}(\rho_t = \rho) \leq \min\left\{ \frac{24\sigma^2 \ln(T)}{(gap(\rho) - 4L\Delta(\rho_t))^2} + 1, \frac{6\sigma^2 \ln(T)}{L^2\Delta^2(\rho)} + 2 \right\} \tag{11}$$

**Lemma B.5.** *Conditioned on the "good events" $\mathcal{G}$, for any $\rho \in \mathcal{P}$ such that $\tau_f(\rho) \leq T$,*

$$\max_{(x,a) \in \rho} (f^*(x) - f_a(x)) \leq 10L\Delta(\rho).$$

*For any $\rho \in \mathcal{P}$ that was created due to a parent in $\mathcal{P}$ being flagged (i.e. $\Delta(\rho) \leq \frac{1}{4}$)*

$$\max_{(x,a) \in \rho} (f^*(x) - f_a(x)) \leq 20L\Delta(\rho).$$

Note that since we assumed the reward functions are bounded in $[0, 1]$, the regret is always bounded by 1, thus the above upper bound is only nontrivial for $\Delta(\rho) < 1/20L$.

**Lemma B.6.** *For any $\rho \in \mathcal{P}$ such that $\tau_f(\rho) \leq T$,*

$$\mathbb{E}\left[\sum_{t=\tau_f(\rho)+1}^{T} \mathbb{I}(\rho_t = \rho) \mid \tau_f(\rho) \leq T\right] \leq \frac{304 \cdot 5431 \cdot \sigma^2 |\mathcal{A}(\rho)| \ln(T|\mathcal{A}(\rho)|)}{L^2 \Delta^2(\rho)} \tag{12}$$

# C  Proof of Lemmas

*Proof of Lemma B.3.* Recall from the algorithm that every ball $\rho \in \mathcal{P}$ is constructed as a result of the SUBPARTITION routine, and is associate to a cluster with a corresponding center arm $y \in \mathcal{A}(\rho)$. Let us denote $r = c_0(\rho)$ and $s = c_1(\rho)$, which are the endpoints used to compute distances within the SUBPARTITION subroutine. It follows that $\Delta(\rho) = s - r$. The algorithm enforced that for any $a, a' \in \mathcal{A}(\rho) \times \mathcal{A}(\rho)$, it must be that $\hat{\mathcal{D}}_r^s(a, y) \leq \frac{3}{16}L(v-u)$ and $\hat{\mathcal{D}}_r^s(a', y) \leq \frac{3}{16}L(v-u)$.

Conditioned on the good events $\mathcal{G}$,

$$|\hat{\mathcal{D}}_u^v(a, y) - \mathcal{D}_u^v(a, y)| \leq \frac{1}{8}L(v-u) \text{ and } |\hat{\mathcal{D}}_u^v(a', y) - \mathcal{D}_u^v(a', y)| \leq \frac{1}{8}L(v-u).$$

We can verify that $\mathcal{D}_s^r(\cdot)$ is indeed a proper metric amongst the arms as it is simply a normalized L2 norm of the difference of associated vectors. Therefore by triangle inequality,

$$\mathcal{D}_u^v(a, a') \leq \mathcal{D}_u^v(a, y) + \mathcal{D}_u^v(a', y) \tag{13}$$

$$\leq \frac{5}{16}L(v-u) + \frac{5}{16}L(v-u) = \frac{5}{8}L(v-u). \tag{14}$$

Recall that

$$\mathcal{D}_u^v(a, a') = \sqrt{\frac{1}{200} \sum_{i \in [200]} \left(f_a(z_i(u, v)) - f_{a'}(z_i(u, v))\right)^2} \text{ for } z_i(u, v) = \left(1 - \frac{i}{200}\right)r + \frac{i}{200}s$$

As $[u, v]$ is a compact set, it must achieve its supremum, and we let $x^*$ denote the point that achieves supremum, i.e.

$$|f_a(x^*) - f_{a'}(x^*)| = \sup_{x \in [u,v]} |f_a(x) - f_{a'}(x)|.$$

As $f_a$ and $f_{a'}$ are $L$-Lipschitz, the absolute value of their difference must be $2L$-Lipschitz.

For $z_i(u, v)$ as defined above, by Lipschitzness,

$$|f_a(z_i(u, v)) - f_{a'}(z_i(u, v))| \geq \max(|f_a(x^*) - f_{a'}(x^*)| - 2L|z_i(u, v) - x^*|, 0)$$

Therefore,

$$(\mathcal{D}_u^v(a, a'))^2 = \frac{1}{200} \sum_{i \in [200]} \left(f_a(z_i(u, v)) - f_{a'}(z_i(u, v))\right)^2$$

$$\geq \frac{1}{200} \sum_{i \in [200]} \left(\max\left(|f_a(x^*) - f_{a'}(x^*)| - 2L|z_i(u, v) - x^*|, 0\right)\right)^2$$

$$\geq \min_{x \in [u,v]} \frac{1}{200} \sum_{i \in [200]} \left(\max\left(|f_a(x^*) - f_{a'}(x^*)| - 2L|z_i(u, v) - x|, 0\right)\right)^2$$

Argue that this is lower bounded by choosing $x = s$ (include a picture?) so that

$$|z_i(u, v) - x| = \left(1 - \frac{i}{200}\right)(v - u)$$

and by rearranging the indices we get

$$\left(\mathcal{D}_u^v(a, a')\right)^2 \geq \frac{1}{200} \sum_{i=0}^{200} \left(\max\left(|f_a(x^*) - f_{a'}(x^*)| - \frac{2L(v - u)i}{200}, 0\right)\right)^2$$

Let us define

$$i_{\max} := \min\left(\left\lfloor \frac{200|f_a(x^*) - f_{a'}(x^*)|}{2L(v - u)} \right\rfloor, 200\right)$$

then we can lower bound $\left(\mathcal{D}_u^v(a, a')\right)^2$ by

$$\frac{1}{200} \sum_{i=0}^{i_{\max}} \left(|f_a(x^*) - f_{a'}(x^*)| - \frac{2L(v - u)i}{200}\right)^2$$

$$= \frac{i_{\max} + 1}{200}|f_a(x^*) - f_{a'}(x^*)|^2 - \frac{2L|f_a(x^*) - f_{a'}(x^*)|(v - u)}{200^2} \sum_{i=0}^{i_{\max}} i + \frac{4L^2(v - u)^2}{200^3} \sum_{i=0}^{i_{\max}} i^2$$

Recall that

$$\sum_{i=0}^{i_{\max}} i = \frac{i_{\max}(i_{\max} + 1)}{2} \leq \frac{(i_{\max} + 1)^2}{2}$$

and

$$\sum_{i=0}^{i_{\max}} i^2 = \frac{i_{\max}(i_{\max} + 1)(2i_{\max} + 1)}{6} \geq \frac{i_{\max}^3}{3}$$

Therefore, if $200 < \left\lfloor \frac{200|f_a(x^*) - f_{a'}(x^*)|}{2L(v - u)} \right\rfloor$,

$$\left(\mathcal{D}_u^v(a, a')\right)^2 \tag{15}$$

$$\geq \frac{200 + 1}{200}|f_a(x^*) - f_{a'}(x^*)|^2 - \frac{2L|f_a(x^*) - f_{a'}(x^*)|(v - u)(200 + 1)^2}{2 \cdot 200^2} + \frac{4L^2(v - u)^2}{3} \tag{16}$$

$$\geq |f_a(x^*) - f_{a'}(x^*)|^2 - \left(\frac{4}{3}\right)^2 L|f_a(x^*) - f_{a'}(x^*)|(v - u) + \frac{4}{3}L^2(v - u)^2 \tag{17}$$

$$= \left(|f_a(x^*) - f_{a'}(x^*)| - \frac{8L(v - u)}{9}\right)^2 + \frac{44}{81}L^2(v - u)^2 \tag{18}$$

We have shown in (14) that conditioned on the good event $\mathcal{G}$, $\mathcal{D}_u^v(a, a') \leq \frac{5}{8}L(v - u)$, which would violate (18), as $\left(\frac{5}{8}\right)^2 \leq \frac{44}{81}$.

Thus, conditioned on the good event $\mathcal{G}$, for a pair of arms $a, a' \in \mathcal{A}(\rho) \times \mathcal{A}(\rho)$, it must be that $200 > \left\lfloor \frac{200|f_a(x^*) - f_{a'}(x^*)|}{2L(v - u)} \right\rfloor$, such that

$$\left(\mathcal{D}_u^v(a, a')\right)^2 \tag{19}$$

$$\geq \frac{1}{200}\left(\frac{|f_a(x^*) - f_{a'}(x^*)|^2 200}{2L(v - u)}\right)|f_a(x^*) - f_{a'}(x^*)| \tag{20}$$

$$- \frac{L|f_a(x^*) - f_{a'}(x^*)|(v - u)}{200^2}\left(\frac{200|f_a(x^*) - f_{a'}(x^*)|}{2L(v - u)} + 1\right)^2 \tag{21}$$

$$+ \frac{4L^2(v - u)^2}{3 \cdot 200^3}\left(\frac{200|f_a(x^*) - f_{a'}(x^*)|}{2L(v - u)} - 1\right)^3 \tag{22}$$

$$= \frac{|f_a(x^*) - f_{a'}(x^*)|^3}{L(v - u)}\left(\frac{1}{2} - \frac{1}{4}\left(1 + \frac{2L(v - u)}{200|f_a(x^*) - f_{a'}(x^*)|}\right)^2 + \frac{1}{6}\left(1 - \frac{2L(v - u)}{200|f_a(x^*) - f_{a'}(x^*)|}\right)^3\right) \tag{23}$$

Suppose that $|f_a(x^*) - f_{a'}(x^*)| > L(v - u)$. We would arise at a contradiction because plugging this bound into (23) would result in

$$(\mathcal{D}_u^v(a, a'))^2 > L^2(v-u)^2 \left( \frac{1}{2} - \frac{1}{4}(1.01)^2 + \frac{1}{6}(0.99)^3 \right) > \left( \frac{5}{8}L(v-u) \right)^2 .$$

Therefore $\mathcal{D}_u^v(a, a') \leq \frac{5}{8}L(v-u)$ must imply that

$$|f_a(x^*) - f_{a'}(x^*)| := \sup_{x \in [c_0(\rho), c_1(\rho)]} |f_a(x) - f_{a'}(x)| \leq L\Delta(\rho).$$

Let us denote $(\overline{x}, \overline{a}) = \operatorname{argmax}_{(x,a) \in \rho} f_a(x)$ and $(\underline{x}, \underline{a}) = \operatorname{argmin}_{(x,a) \in \rho} f_a(x)$. Then

$$diam(\rho) = f_{\overline{a}}(\overline{x}) - f_{\underline{a}}(\underline{x}) \tag{24}$$
$$\leq |f_{\overline{a}}(\overline{x}) - f_{\overline{a}}(\underline{x})| + |f_{\overline{a}}(\underline{x}) - f_{\underline{a}}(\underline{x})| \tag{25}$$
$$\leq L\Delta(\rho) + \max_{a,a' \in \mathcal{A}(\rho)^2} \sup_{x \in [c_0(\rho), c_1(\rho)]} |f_a(x) - f_{a'}(x)| \tag{26}$$
$$\leq 2L\Delta(\rho) \tag{27}$$

$$\square$$

*Proof of Lemma B.4.* Recall that a ball is flagged (after being played) the first trial that

$$n_t(\rho) > \frac{6\sigma^2 \ln(T)}{L^2 \Delta^2(\rho)},$$

and subsequently it is removed from $\mathcal{P}$ and no longer active, so the flagging condition is triggered exactly when

$$n_t(\rho) = \left\lfloor \frac{6\sigma^2 \ln(T)}{L^2 \Delta^2(\rho)} + 1 \right\rfloor .$$

Since the ball is played one last trial, the total number of trials the ball is played via the UCB rule over time horizon $T$ will be

$$\sum_{t=1}^{\tau_f(\rho)} \mathbb{I}(\rho_t = \rho) = \left\lfloor \frac{6\sigma^2 \ln(T)}{L^2 \Delta^2(\rho)} + 1 \right\rfloor + 1 \leq \frac{6\sigma^2 \ln(T)}{L^2 \Delta^2(\rho)} + 2.$$

The second terms in the upper bound of Lemma B.4 are derived by considering when the ball must be "flagged".

Next, we consider when the ball must be so suboptimal such that it is no longer chosen by the UCB rule. Conditioned on the "good events" $\mathcal{G}$, for all $\rho \in \mathcal{P}$ and $t \in [\tau_f(\rho)]$, i.e. the ball is not yet flagged,

$$\mu_t(\rho) \in \left[ \min_{(x,a) \in \rho} f_a(x) - \sqrt{\frac{6\sigma^2 \ln(T)}{n_t(\rho)}}, \ \max_{(x,a) \in \rho} f_a(x) + \sqrt{\frac{6\sigma^2 \ln(T)}{n_t(\rho)}} \right]$$

such that

$$UCB_t(\rho) \in \left[ \min_{(x,a) \in \rho} f_a(x) + 2L\Delta(\rho), \ \max_{(x,a) \in \rho} f_a(x) + 2L\Delta(\rho) + 2\sqrt{\frac{6\sigma^2 \ln(T)}{n_t(\rho)}} \right].$$

Thus

$$UCB_t(\rho) \geq \min_{(x,a) \in \rho} f_a(x) + 2L\Delta(\rho) = \max_{(x,a) \in \rho} f_a(x) - diam(\rho) + 2L\Delta(\rho)$$

and

$$UCB_t(\rho) \leq \max_{(x,a) \in \rho} f_a(x) + 2L\Delta(\rho) + 2\sqrt{\frac{6\sigma^2 \ln(T)}{n_t(\rho)}} = \min_{(x,a) \in \rho} f_a(x) + diam(\rho) + 2L\Delta(\rho) + 2\sqrt{\frac{6\sigma^2 \ln(T)}{n_t(\rho)}}.$$

Suppose at trial $t$, context $x_t$ arrives and ball $\rho_t$ is chosen by the algorithm via the UCB rule. Let us denote $a^* = \operatorname{argmax}_{a \in [K]} f_a(x_t)$, and let $\rho^*$ denote the ball which contains $a^*$. Then

$$UCB_t(\rho^*) \geq f^*(x_t) - diam(\rho^*) + 2L\Delta(\rho^*).$$

By Lemma B.3, conditioned on the good events $\mathcal{G}$, $diam(\rho^*) \leq 2L\Delta(\rho^*)$, thus

$$UCB_t(\rho^*) \geq f^*(x_t).$$

By conditioning on the good events $\mathcal{G}$, and by Lemma B.3, it also must hold that

$$UCB_t(\rho_t) \leq \min_{(x,a)\in\rho_t} f_a(x) + diam(\rho_t) + 2L\Delta(\rho_t) + 2\sqrt{\frac{6\sigma^2 \ln(T)}{n_t(\rho_t)}} \quad (28)$$

$$\leq \min_{(x,a)\in\rho_t} f_a(x) + 4L\Delta(\rho_t) + 2\sqrt{\frac{6\sigma^2 \ln(T)}{n_t(\rho_t)}}. \quad (29)$$

Thus,

$$\min_{a\in\mathcal{A}(\rho_t)} f_a(x_t) \geq UCB_t(\rho_t) - 4L\Delta(\rho_t) - 2\sqrt{\frac{6\sigma^2 \ln(T)}{n_t(\rho_t)}}.$$

If $\rho_t$ was chosen by the algorithm via the UCB rule, it therefore must imply that

$$UCB_t(\rho) \geq UCB_t(\rho^*)$$

and thus

$$f^*(x_t) - \min_{a\in\mathcal{A}(\rho_t)} f_a(x_t) \leq 4L\Delta(\rho_t) + 2\sqrt{\frac{6\sigma^2 \ln(T)}{n_t(\rho_t)}}.$$

This implies that

$$gap(\rho) = \min_{(x,a)\in\rho} (f^*(x) - f_a(x)) \leq 4L\Delta(\rho_t) + 2\sqrt{\frac{6\sigma^2 \ln(T)}{n_t(\rho_t)}}$$

Therefore if $\rho$ is selected by the UCB rule at trial $t$, it must be that

$$n_t(\rho_t) \leq \frac{24\sigma^2 \ln(T)}{(gap(\rho) - 4L\Delta(\rho_t))^2}.$$

$\square$

*Proof of Lemma B.5.* Note that $\max_{a\in\mathcal{A}(\rho)}(f^*(x) - f_a(x))$ is a $2L$-Lipschitz function such that

$$\max_{x\in[c_0(\rho),c_1(\rho)]} \max_{a\in\mathcal{A}(\rho)} (f^*(x) - f_a(x)) \quad (30)$$

$$\leq \min_{x\in[c_0(\rho),c_1(\rho)]} \max_{a\in\mathcal{A}(\rho)} (f^*(x) - f_a(x)) + 2L\Delta(\rho) \quad (31)$$

$$\leq \min_{x\in[c_0(\rho),c_1(\rho)]} \min_{a\in\mathcal{A}(\rho)} (f^*(x) - f_a(x)) + diam(\rho) + 2L\Delta(\rho) \quad (32)$$

$$= gap(\rho) + diam(\rho) + 2L\Delta(\rho). \quad (33)$$

Conditioned on the "good events" holding, by Lemma B.4, if a ball $\rho$ is flagged at trial $t = \tau_f(\rho)$, then it must be that

$$n_t(\rho) = \left\lfloor \frac{6\sigma^2 \ln(T)}{L^2\Delta^2(\rho)} + 1 \right\rfloor \leq \frac{24\sigma^2 \ln(T)}{(gap(\rho) - 4L\Delta(\rho))^2}. \quad (34)$$

If the above inequality does not hold, then it would imply that the ball is so suboptimal that it stop being chosen by the UCB rule before hitting the threshold for flagging.

This implies

$$\frac{6\sigma^2 \ln(T)}{L^2\Delta^2(\rho)} \leq \frac{24\sigma^2 \ln(T)}{(gap(\rho) - 4L\Delta(\rho))^2} \quad (35)$$

and thus

$$gap(\rho) \leq 6L\Delta(\rho). \quad (36)$$

By Lemma B.3 and (33), it follows that

$$\max_{(x,a)\in\rho} (f^*(x) - f_a(x)) \le gap(\rho) + diam(\rho) + 2L\Delta(\rho)$$
$$\le 6L\Delta(\rho) + 2L\Delta(\rho) + 2L\Delta(\rho)$$
$$= 10L\Delta(\rho).$$

This implies that the maximum regret for a ball who is eventually flagged is upper bounded by $10L\Delta(\rho)$. For any subsequent children balls $\rho'$ that are formed by subpartitioning $\rho$, it must follow that

$$gap(\rho') \le \max_{(x,a)\in\rho} (f^*(x) - f_a(x)) \tag{37}$$
$$\le 10L\Delta(\rho) \tag{38}$$
$$= 20L\Delta(\rho') \tag{39}$$

$\square$

*Proof of Lemma B.6.* The length of trial that a ball $\rho$ stays in the flagged phase before triggering the SUBPARTITION subroutine depends on the number of samples collected for this ball until the condition $\prod_{a\in\mathcal{A}(\rho)}$ SUFFDATA$(a) == 1$. Essentially this condition considers the 64 equal sized intervals that split the context space $[c_0(\rho), c_1(\rho)]$ and checks that in each there is at least $k$ points sampled for each arm $a \in \mathcal{A}(\rho)$ within each of those 64 subintervals.

Due to the fact that the algorithm always gives priority to flagged balls, and furthermore flagged balls with larger context width are always given priority, it follows that the context of samples collected for ball $\rho$ within its flagged phase must be distributed uniformly within $[c_0(\rho), c_1(\rho)]$. If $\rho$ is the only ball flagged that intersects with the context interval $[c_0(\rho), c_1(\rho)]$, then it is given first priority such that any context that falls within this interval will be assigned to $\rho$. As the contexts arrive uniformly sampled over $[0, 1]$, the set of contexts restricted to $[c_0(\rho), c_1(\rho)]$ will also be uniformly distributed.

As the context widths are always split into half, the endpoints must be equal to $\ell 2^{-i}$ for some integers $i$ and $\ell$. In the case that there is some other ball $\rho'$ which intersects with $[c_0(\rho), c_1(\rho)]$, either it has smaller context width and thus must be fully contained within $[c_0(\rho), c_1(\rho)]$, or it has larger context width and must be a strict superset fully eclipsing $[c_0(\rho), c_1(\rho)]$. It is impossible for there to be another flagged ball $\rho'$ with the exact same context width $[c_0(\rho), c_1(\rho)]$, because whichever ball was flagged first, would cause the algorithm to give full priority to that flagged ball, so that it would be impossible while that ball is still flagged, for any context in $[c_0(\rho), c_1(\rho)]$ to be assigned to the other ball to trigger it to be flagged. If $\rho'$ has smaller width, then $\rho'$ is given lower priority, and no samples will be assigned to $\rho'$ until $\rho$ is subpartitioned and unflagged. If $\rho'$ has larger width, then $\rho'$ will be given higher priority over the entire interval $[c_0(\rho), c_1(\rho)]$, such that no samples will be assigned to $\rho$ until $\rho'$ is subpartitioned and unflagged.

As a result, in each trial $t$, either $\rho$ has priority on the entire context interval $[c_0(\rho), c_1(\rho)]$ and thus receives samples uniformly within that interval, or $\rho$ does not have priority on any subset of the interval $[c_0(\rho), c_1(\rho)]$ such that it receives no samples at all, guarantees that the eventual set of samples collected during the flagged phase must be distributed uniformly on $[c_0(\rho), c_1(\rho)]$. As a result, the probability that each sample collected for $\rho$ falls into any of the 64 subintervals of $[c_0(\rho), c_1(\rho)]$ is evenly 1/64. By coupon collector, the expected number of samples until we get one sample in each bucket is $64\sum_{i=1}^{64}\frac{1}{i}$. A naive upper bound on the number of samples until we get $k$ samples in each bucket is $64k\sum_{i=1}^{64}\frac{1}{i} \le 304k$.

As there are $|\mathcal{A}(\rho)|$ arms, each of which needs to satisfy SUFFDATA$(a) == 1$,

$$\mathbb{E}\left[\sum_{t=\tau_f(\rho)+1}^{T} \mathbb{I}(\rho_t = \rho) \mid \tau_f(\rho) \le T\right] \le 304k|\mathcal{A}(\rho)| = \frac{304 \cdot 5431 \cdot \sigma^2|\mathcal{A}(\rho)|\ln(T|\mathcal{A}(\rho)|)}{L^2\Delta^2(\rho)} \tag{40}$$

$\square$

# D Bounding Probability of Bad Events

*Proof of Lemma B.1.* Recall that by definition,

$$\mu_t(\rho) = \frac{1}{n_t(\rho)} \sum_{s=1}^{t-1} \mathbb{I}(\rho_s = \rho) \pi_s$$

$$= \frac{1}{n_t(\rho)} \sum_{s=1}^{t-1} \mathbb{I}(\rho_s = \rho) (f_{a_s}(x_s) + \epsilon_s)$$

As $(x_s, a_s) \in \rho$ if $\rho_s = \rho$, it follows that

$$\frac{1}{n_t(\rho)} \sum_{s=1}^{t-1} \mathbb{I}(\rho_s = \rho) f_{a_s}(x_s) \in \left[ \min_{(x,a)\in\rho} f_a(x), \max_{(x,a)\in\rho} f_a(x) \right].$$

It remains to be shown that with high probability, for all $t$,

$$\left| \frac{1}{n_t(\rho)} \sum_{s=1}^{t-1} \mathbb{I}(\rho_s = \rho) \epsilon_s \right| \leq \sqrt{\frac{6\sigma^2 \ln(T)}{n_t(\rho)}}.$$

Note that we really only need to concern ourselves with values of $t$ after which the ball has been chosen at least once. Otherwise, before the ball has been chosen yet, there are no terms to sum over, thus trivially zero is bounded above by the confidence bound.

For some ball $\rho$, let us denote the sequence

$$(\tau_1, \tau_2, \tau_3, ...)$$

where $\tau_s$ corresponds to the $s$-th trial that ball $\rho$ is chosen by the algorithm, i.e.

$$\sum_{t=1}^{\tau_s} \mathbb{I}(\rho_t = \rho) = s \text{ and } \rho_{\tau_s} = \rho.$$

By Doob's optional skipping theorem,

$$(\epsilon_{\tau_1}, \epsilon_{\tau_2}, \epsilon_{\tau_3}, ...)$$

is identically distributed to

$$(\epsilon_1, \epsilon_2, \epsilon_3, ...).$$

Therefore, by Doob's optional skipping theorem, union bound, and Hoeffding's inequality,

$$\mathbb{P}(\forall s \in [T], \left| \frac{1}{s} \sum_{\ell=1}^{s} \epsilon_{\tau_\ell} \right| \leq \sqrt{\frac{6\sigma^2 \ln(T)}{s}})$$

$$= \mathbb{P}(\forall s \in [T], \left| \frac{1}{s} \sum_{\ell=1}^{s} \epsilon_\ell \right| \leq \sqrt{\frac{6\sigma^2 \ln(T)}{s}})$$

$$\leq \sum_{s\in[T]} \mathbb{P}(\left| \sum_{\ell=1}^{s} \epsilon_\ell \right| \leq \sqrt{\frac{6s^2\sigma^2 \ln(T)}{s}})$$

$$\leq \sum_{s\in[T]} 2 \exp(-3 \ln(T))$$

$$\leq 2T^{-2}$$

There are at most $T$ active balls over the course of the algorithm, thus by union bound over all active balls $\rho$ over the course of the algorithm, with probability at least $1 - 2T^{-1}$, for all active balls $\rho$, for all $t : n_t(\rho) \geq 1$,

$$\mu_t(\rho) \in \left[ \min_{(x,a)\in\rho} f_a(x) - \sqrt{\frac{6\sigma^2 \ln(T)}{n_t(\rho)}}, \max_{(x,a)\in\rho} f_a(x) + \sqrt{\frac{6\sigma^2 \ln(T)}{n_t(\rho)}} \right].$$

$\square$

*Proof of Lemma B.2.* Lemma D.1 proves that each trial $\hat{\mathcal{D}}_u^v(a, y)$ is evaluated within the subroutine SUBPARTITION$(\rho)$ over the course of the algorithm,

$$\left\{ |\hat{\mathcal{D}}_u^v(a, y) - \mathcal{D}_u^v(a, y)| \leq \frac{1}{8} L(v - u) \right\}$$

with probability at least $1 - 4T^{-2}|\mathcal{A}(\rho)|^{-2}$.

Within a subroutine SUBPARTITION$(\rho)$, the maximum number of trials that $\hat{\mathcal{D}}_u^v(a, y)$ could be evaluated is $|\mathcal{A}(\rho)|^{-2}$. There are maximally $T$ trials that subroutine SUBPARTITION can be called. By union bound, with probability $1 - 4T^{-1}$, over the entire course of the algorithm, for all trials that $\hat{\mathcal{D}}_u^v(a, y)$ is evaluated,

$$\left\{ |\hat{\mathcal{D}}_u^v(a, y) - \mathcal{D}_u^v(a, y)| \leq \frac{1}{8} L(v - u) \right\}.$$

$\square$

**Lemma D.1.** *Each trial $\hat{\mathcal{D}}_u^v(a, y)$ is evaluated within the subroutine* SUBPARTITION *over the course of the algorithm,*

$$\left\{ |\hat{\mathcal{D}}_u^v(a, y) - \mathcal{D}_u^v(a, y)| \leq \frac{1}{8} L(v - u) \right\}$$

*with probability at least $1 - 4T^{-2}|\mathcal{A}(\rho)|^{-2}$.*

*Proof.* Consider a single call to the subroutine SUBPARTITION for a ball $\rho$. By construction, the subroutine is only called when $\prod_{a \in \mathcal{A}(\rho)}$ SUFFDATA$(a) == 1$, where

$$\text{SUFFDATA}(a) = \prod_{i=1}^{64} \mathbb{I} \left( \sum_{s > \tau_f(\rho)} \mathbb{I}(\rho_s = \rho, a_s = a) \, \mathbb{I}(x_s \in [w_{i-1}, w_i]) \geq k \right) \quad (41)$$

for $w_i = c_0(\rho) + \frac{i\Delta(\rho)}{64}$. As a result, by construction, each trial $\hat{\mathcal{D}}_u^v(a, y)$ is evaluated within the subroutine SUBPARTITION, if we take the interval $[u, v]$ and split it evenly into 32 subintervals, the algorithm guarantees that for each of the 32 subintervals, for each arm $a$ and $y$, there are at least $k$ samples (or observations) collected for $\rho$ during the "flagged phase" such that the context lies within the subinterval. As our algorithm estimates the reward functions $\hat{f}_a$ and $\hat{f}_y$ via $k$ nearest neighbor averaging, this condition guarantees a minimum bias.

Recall that

$$\hat{\mathcal{D}}_u^v(a, y) := \sqrt{\frac{1}{200} \sum_{i \in [200]} \left( \hat{f}_a(z_i(u, v)) - \hat{f}_y(z_i(u, v)) \right)^2 - \frac{2\sigma^2}{k}} \quad (42)$$

for

$$\hat{f}_a(x) = \frac{1}{k} \sum_{s = \tau_f(\rho)+1}^{\tau_{cl}(\rho)} \mathbb{I}(\rho_s = \rho, a_s = a) \, \mathbb{I}(x_s \in \text{k-NN}) \, \pi_s, \quad (43)$$

where $x_s$ is a $k$ nearest neighbor datapoint for computing $\hat{f}_a(x)$ if

$$\mathbb{I}(x_s \in \text{k-NN}) = \sum_{\ell = \tau_f(\rho)+1}^{\tau_{cl}(\rho)} \mathbb{I}(\rho_\ell = \rho, a_\ell = a) \, \mathbb{I}(|x_\ell - x| \leq |x_s - x|) \leq k.$$

Recall that $\pi_s = f_{a_s}(x_s) + \epsilon_s$, where $\epsilon_s$ is an independent noise term distributed as $N(0, \sigma^2)$. Let us denote

$$\bar{\mathcal{D}}_u^v(a, y) := \sqrt{\frac{1}{200} \sum_{i \in [200]} \left( \bar{f}_a(z_i(u, v)) - \bar{f}_y(z_i(u, v)) \right)^2} \quad (44)$$

for

$$\bar{f}_a(x) = \frac{1}{k} \sum_{s = \tau_f(\rho)+1}^{\tau_{cl}(\rho)} \mathbb{I}(\rho_s = \rho, a_s = a) \, \mathbb{I}(x_s \in \text{k-NN}) \, f_a(x_s). \quad (45)$$

As a result of the condition $\prod_{a \in \mathcal{A}(\rho)}$ SUFFDATA$(a) == 1$, for any $x$, if $x_s \in$ k-NN, then $x_s - x \leq \frac{v-u}{32}$, such that $|f_a(x) - f_a(x_s)| \leq \frac{L(v-u)}{32}$. This implies that $|f_a(x) - \bar{f}_a(x)| \leq \frac{L(v-u)}{32}$

By triangle inequality and Lipschitzness,

$$|\mathcal{D}_u^v(a,y) - \bar{\mathcal{D}}_u^v(a,y)|$$

$$\leq \sqrt{\frac{1}{200}\sum_{i\in[200]}\left((f_a(z_i(u,v)) - f_y(z_i(u,v))) - (\bar{f}_a(z_i(u,v)) - \bar{f}_y(z_i(u,v)))\right)^2}$$

$$\leq \sqrt{\frac{1}{200}\sum_{i\in[200]}\left(\frac{L(v-u)}{16}\right)^2}$$

$$\leq \frac{L(v-u)}{16}.$$

By Lemma D.2, with probability at least $1 - 2T^{-2}|\mathcal{A}(\rho)|^{-2}$, $|\bar{\mathcal{D}}_u^v(a,y) - \hat{\mathcal{D}}_u^v(a,y)| \leq \frac{L(v-u)}{16}$. It follows that

$$|\mathcal{D}_u^v(a,y) - \hat{\mathcal{D}}_u^v(a,y)| \leq |\mathcal{D}_u^v(a,y) - \bar{\mathcal{D}}_u^v(a,y)| + |\bar{\mathcal{D}}_u^v(a,y) - \hat{\mathcal{D}}_u^v(a,y)| \leq \frac{L(v-u)}{8}.$$

$\square$

**Lemma D.2.** *With probability at least* $1 - 4T^{-2}|\mathcal{A}(\rho)|^{-2}$,

$$|\hat{\mathcal{D}}_u^v(a,y) - \bar{\mathcal{D}}_u^v(a,y)| \leq \frac{L(v-u)}{16}.$$

*Proof.* Let us denote $\mathcal{E}(\rho,a,i) = \{t \in [\tau_f(\rho)+1, \tau_{cl}(\rho)] \ s.t. \ \rho_t = \rho, a_t = a, x_t \in$ k-NN of $z_i(u,v)\}$. Recall that by definition,

$$\hat{f}_a(z_i(u,v)) = \bar{f}_a(z_i(u,v)) + \frac{1}{k}\sum_{t\in\mathcal{E}(\rho,a,i)}\epsilon_t$$

We split $|(\hat{\mathcal{D}}_u^v(a,y))^2 - (\bar{\mathcal{D}}_u^v(a,y))^2|$ into two terms,

$$|(\hat{\mathcal{D}}_u^v(a,y))^2 - (\bar{\mathcal{D}}_u^v(a,y))^2| \tag{46}$$

$$= |\frac{1}{200}\sum_{i\in[200]}\left((\hat{f}_a(z_i(u,v)) - \hat{f}_y(z_i(u,v)))^2 - (\bar{f}_a(z_i(u,v)) - \bar{f}_y(z_i(u,v)))^2\right) - \frac{2\sigma^2}{k}| \tag{47}$$

$$\leq |\frac{2}{200}\sum_{i\in[200]}(\bar{f}_a(z_i(u,v)) - \bar{f}_y(z_i(u,v)))\left(\frac{1}{k}\sum_{t\in\mathcal{E}(\rho,a,i)}\epsilon_t - \frac{1}{k}\sum_{t\in\mathcal{E}(\rho,y,i)}\epsilon_t\right)| \tag{48}$$

$$+ |\frac{1}{200}\sum_{i\in[200]}\left(\frac{1}{k}\sum_{t\in\mathcal{E}(\rho,a,i)}\epsilon_t - \frac{1}{k}\sum_{t\in\mathcal{E}(\rho,y,i)}\epsilon_t\right)^2 - \frac{2\sigma^2}{k}| \tag{49}$$

By Lemmas D.3 and D.4, with probability at least $1 - 4T^{-2}|\mathcal{A}(\rho)|^{-2}$,

$$|(\hat{\mathcal{D}}_u^v(a,y))^2 - (\bar{\mathcal{D}}_u^v(a,y))^2| \leq \bar{\mathcal{D}}_u^v(a,y)\sqrt{\frac{4\sigma^2\ln(T|\mathcal{A}(\rho)|)}{k}} + \frac{4\sigma^2\ln(T|\mathcal{A}(\rho)|)}{k}.$$

Let us denote $Q_1 = \sqrt{\frac{4\sigma^2\ln(T|\mathcal{A}(\rho)|)}{k}}$.

We consider two bounds for $|\bar{\mathcal{D}}_u^v(a,y) - \hat{\mathcal{D}}_u^v(a,y)|$,

$$|\hat{\mathcal{D}}_u^v(a,y) - \bar{\mathcal{D}}_u^v(a,y)| \leq \hat{\mathcal{D}}_u^v(a,y) + \bar{\mathcal{D}}_u^v(a,y)$$

$$\leq \sqrt{|(\hat{\mathcal{D}}_u^v(a,y))^2 - (\bar{\mathcal{D}}_u^v(a,y))^2| + (\bar{\mathcal{D}}_u^v(a,y))^2} + \bar{\mathcal{D}}_u^v(a,y)$$

$$\leq \sqrt{Q_1(\bar{\mathcal{D}}_u^v(a,y) + Q_1) + (\bar{\mathcal{D}}_u^v(a,y))^2} + \bar{\mathcal{D}}_u^v(a,y)$$

and

$$|\hat{\mathcal{D}}_u^v(a,y) - \bar{\mathcal{D}}_u^v(a,y)| \leq \frac{|(\hat{\mathcal{D}}_u^v(a,y))^2 - (\bar{\mathcal{D}}_u^v(a,y))^2|}{\hat{\mathcal{D}}_u^v(a,y) + \bar{\mathcal{D}}_u^v(a,y)}$$

$$\leq \frac{|(\hat{\mathcal{D}}_u^v(a,y))^2 - (\bar{\mathcal{D}}_u^v(a,y))^2|}{\bar{\mathcal{D}}_u^v(a,y)}$$

$$\leq \frac{Q_1(\bar{\mathcal{D}}_u^v(a,y) + Q_1)}{\bar{\mathcal{D}}_u^v(a,y)}.$$

The first bound is strictly increasing in $\bar{\mathcal{D}}_u^v(a,y)$, whereas the second bound is strictly decreasing in $\bar{\mathcal{D}}_u^v(a,y)$. Therefore for any $Q_2$,

$$|\hat{\mathcal{D}}_u^v(a,y) - \bar{\mathcal{D}}_u^v(a,y)| \leq \min\left(\bar{\mathcal{D}}_u^v(a,y) + \sqrt{Q_1(\bar{\mathcal{D}}_u^v(a,y) + Q_1) + (\bar{\mathcal{D}}_u^v(a,y))^2}, Q_1 + \frac{Q_1^2}{\bar{\mathcal{D}}_u^v(a,y)}\right)$$

$$\leq \max\left(Q_2 + \sqrt{Q_1 Q_2 + Q_1^2 + Q_2^2}, Q_1 + \frac{Q_1^2}{Q_2}\right)$$

Let us choose $Q_2 = \frac{1+\sqrt{13}}{6} Q_1$ so that

$$|\hat{\mathcal{D}}_u^v(a,y) - \bar{\mathcal{D}}_u^v(a,y)| \leq \frac{7+\sqrt{13}}{1+\sqrt{13}} Q_1$$

$$\leq \frac{7+\sqrt{13}}{1+\sqrt{13}} \sqrt{\frac{4\sigma^2 \ln(T|\mathcal{A}(\rho)|)}{k}}$$

For

$$k = \frac{5431 \sigma^2 \ln(T|\mathcal{A}(\rho)|)}{L^2(v-u)^2} \geq 16^2 \left(\frac{7+\sqrt{13}}{1+\sqrt{13}}\right)^2 \frac{4\sigma^2 \ln(T|\mathcal{A}(\rho)|)}{L^2(v-u)^2},$$

it follows that

$$|\hat{\mathcal{D}}_u^v(a,y) - \bar{\mathcal{D}}_u^v(a,y)| \leq \frac{L(v-u)}{16}.$$

$\square$

**Lemma D.3.** *With probability at least* $1 - 2T^{-2}|\mathcal{A}(\rho)|^{-2}$,

$$(48) \leq \bar{\mathcal{D}}_u^v(a,y) \sqrt{\frac{4\sigma^2 \ln(T|\mathcal{A}(\rho)|)}{k}}.$$

*Proof.* Note that $\frac{2}{200} \sum_{i \in [200]} (\bar{f}_a(z_i(u,v)) - \bar{f}_y(z_i(u,v)))\left(\frac{1}{k} \sum_{t \in \mathcal{E}(\rho,a,i)} \epsilon_t - \frac{1}{k} \sum_{t \in \mathcal{E}(\rho,y,i)} \epsilon_t\right)$ is simply a mean zero Gaussian random variable, thus we only need to compute the variance and then we can apply Hoefdding's Inequality.

Let us denote

$$Y_i = (\bar{f}_a(z_i(u,v)) - \bar{f}_y(z_i(u,v)))\left(\frac{1}{k} \sum_{t \in \mathcal{E}(\rho,a,i)} \epsilon_t - \frac{1}{k} \sum_{t \in \mathcal{E}(\rho,y,i)} \epsilon_t\right),$$

such that the expression of interest to us can be restated as $\frac{2}{200} \sum_{i \in [200]} Y_i$. Note that

$$\text{Var}[Y_i] = (\bar{f}_a(z_i(u,v)) - \bar{f}_y(z_i(u,v)))^2 \frac{2\sigma^2}{k}$$

Then

$$\text{Var}\left[\frac{2}{200} \sum_{i \in [200]} Y_i\right] = \frac{4}{200^2} \sum_{i,i' \in [200]^2} \text{Cov}(Y_i, Y_{i'})$$

By construction, $\text{Cov}(Y_i, Y_{i'})$ is zero if $|z_i(u,v) - z_{i'}(u,v)| > \frac{v-u}{16}$, as all the $k - NN$ points must be within distance $\frac{v-u}{32}$. Additionally,

$$\text{Cov}(Y_i, Y_i') \leq \frac{1}{2}(\text{Var}[Y_i] + \text{Var}[Y_{i'}]).$$

Then

$$\frac{4}{200^2} \sum_{i,i' \in [200]^2} \mathrm{Cov}(Y_i, Y_i')$$

$$\leq \frac{4}{200^2} \sum_{i,i' \in [200]^2} \tfrac{1}{2}(\mathrm{Var}[Y_i] + \mathrm{Var}[Y_{i'}]) \mathbb{I}\left(|z_i(u,v) - z_{i'}(u,v)| \leq \tfrac{v-u}{16}\right)$$

$$\leq \frac{4}{200^2} \sum_{i \in [200]} \mathrm{Var}[Y_i] \sum_{i' \in [200]} \mathbb{I}\left(|z_i(u,v) - z_{i'}(u,v)| \leq \tfrac{v-u}{16}\right)$$

$$\leq \frac{4}{200^2} \sum_{i \in [200]} \mathrm{Var}[Y_i] \max(1, \tfrac{200}{8})$$

$$\leq \frac{\sigma^2}{200k} \sum_{i \in [200]} (\bar{f}_a(z_i(u,v)) - \bar{f}_y(z_i(u,v)))^2$$

$$= \frac{\sigma^2}{k} (\bar{\mathcal{D}}_u^v(a,y))^2$$

From Hoeffding's inequality,

$$P[(48) \geq \delta] \leq 2 \exp\left\{ -\frac{\delta^2 k}{2\sigma^2 (\bar{\mathcal{D}}_u^v(a,y))^2} \right\} \tag{50}$$

For $\delta = \bar{\mathcal{D}}_u^v(a,y) \sqrt{\frac{4\sigma^2 \ln(T|\mathcal{A}(\rho)|)}{k}}$, with probability at least $1 - 2T^{-2}|\mathcal{A}(\rho)|^{-2}$, $(48) < \delta$. $\qquad\square$

**Lemma D.4.** *With probability at least* $1 - 2T^{-2}|\mathcal{A}(\rho)|^{-2}$,

$$(49) \leq \frac{4\sigma^2 \ln(T|\mathcal{A}(\rho)|)}{k}.$$

*Proof.* Let us define
$$Y_i = \tfrac{1}{k} \sum_{t \in \mathcal{E}(\rho,a,i)} \epsilon_t - \tfrac{1}{k} \sum_{t \in \mathcal{E}(\rho,y,i)} \epsilon_t,$$

so that the expression in (49) can be rewritten as $\left|\frac{1}{200} \sum_{i \in [200]} Y_i^2 - \frac{2\sigma^2}{k}\right|$. The left and right terms of $Y_i$ are independent as they correspond to samples obtained for different arms, $a$ and $y$, so $\mathcal{E}(\rho,a,i)$ is completely disjoint from $\mathcal{E}(\rho,y,i)$. Therefore $Y_i \sim N(0, \frac{2\sigma^2}{k})$ such that $\mathbb{E}\left[\frac{1}{200} \sum_{i \in [200]} Y_i^2\right] = \frac{2\sigma^2}{k}$.

Next, we want to show that $\frac{1}{200} \sum_{i \in [200]} Y_i^2$ is sub-exponential, so that we can use Bernstein's inequality to bound its concentration around its mean. The vector $(Y_i)_{i \in [200]}$ can be written as a affine transformation $QX$, where $Q$ is the matrix defined as

$$Q_{it} = \tfrac{1}{k}(\mathbb{I}(t \in \mathcal{E}(\rho,a,i)) - \mathbb{I}(t \in \mathcal{E}(\rho,y,i))),$$

and $X_t = \epsilon_t$, such that $X \sim N(0, \sigma^2 I)$. Therefore the vector $(Y_i)_{i \in [200]}$ is a multivariate Gaussian with mean zero and variance $\sigma^2 QQ^T$.

It holds by the sub-exponential property of sum of Gaussians that conditioned on the latent variables $\{\beta_i\}_{i \in [m]}$ and the observation indices $\mathcal{E}'$,

$$\mathbb{P}\left(\left|\frac{Y^T Y}{200} - \frac{\|\sigma^2 QQ^T\|_*}{200}\right| \geq \delta\right) \leq \begin{cases} 2\exp\left(-\frac{200^2 \delta^2}{8\|\sigma^2 QQ^T\|_F^2}\right) & \text{if } \delta \leq \frac{\|\sigma^2 QQ^T\|_F^2}{200\|\sigma^2 QQ^T\|_2} \\ 2\exp\left(-\frac{200\delta}{8\|\sigma^2 QQ^T\|_2}\right) & \text{if } \delta > \frac{\|\sigma^2 QQ^T\|_F^2}{200\|\sigma^2 QQ^T\|_2} \end{cases}.$$

$\|\cdot\|_*$ denotes the nuclear norm, $\|\cdot\|_F$ denotes the frobenius norm, and $\|\cdot\|_2$ denotes the spectral norm. We can verify that indeed

$$\frac{\|\sigma^2 QQ^T\|_*}{200} = \frac{\sigma^2 Tr(QQ^T)|}{200} = \frac{2\sigma^2}{k}.$$

Let us first upper bound $|[QQ^T]_{ij}|$,

$$|[QQ^T]_{ij}| = |\sum_t Q_{it}Q_{jt}|$$

$$= |\frac{1}{k^2}\sum_t (\mathbb{I}(t \in \mathcal{E}(\rho,a,i)) - \mathbb{I}(t \in \mathcal{E}(\rho,y,i)))(\mathbb{I}(t \in \mathcal{E}(\rho,a,j)) - \mathbb{I}(t \in \mathcal{E}(\rho,y,j)))|$$

$$\leq \mathbb{I}\left(|z_i(u,v) - z_j(u,v)| \leq \frac{v-u}{16}\right)\frac{2}{k}.$$

We use this to upper bound $\|\sigma^2 QQ^T\|_F^2$,

$$\|\sigma^2 QQ^T\|_F^2 = \sum_{i \in 200}\sum_{j \in 200}([\sigma^2 QQ^T]_{ij})^2$$

$$\leq \sigma^4 \sum_{i \in 200}\sum_{j \in 200}\mathbb{I}\left(|z_i(u,v) - z_j(u,v)| \leq \frac{v-u}{16}\right)\frac{4}{k^2}$$

$$\leq \frac{4\sigma^4}{k^2}\sum_{i \in 200}\frac{200}{8}$$

$$= \frac{200^2 \sigma^4}{2k^2}$$

By symmetry, $\|\sigma^2 QQ^T\|_1 = \|\sigma^2 QQ^T\|_\infty$. By Holder's inequality,

$$\|\sigma^2 QQ^T\|_2 \leq \sqrt{\|\sigma^2 QQ^T\|_1\|\sigma^2 QQ^T\|_\infty} \tag{51}$$

$$= \max_{j \in [200]}\sum_{i \in [200]}|[\sigma^2 QQ^T]_{ij}| \tag{52}$$

$$\leq \frac{2\sigma^2}{k}\max_{j \in [200]}\sum_{i \in [200]}\mathbb{I}\left(|z_i(u,v) - z_j(u,v)| \leq \frac{v-u}{16}\right) \tag{53}$$

$$\leq \frac{2\sigma^2}{k}\max_{j \in [200]}\frac{200}{8} \tag{54}$$

$$\leq \frac{200\sigma^2}{4k} \tag{55}$$

By plugging these bounds in, we obtain that

$$\mathbb{P}\left(\left|\frac{Y^T Y}{200} - \frac{2\sigma^2}{k}\right| \geq \delta\right) \leq \begin{cases} 2\exp\left(-\frac{\delta^2 k^2}{4\sigma^4}\right) & \text{if } \delta \leq \frac{2\sigma^2}{k} \\ 2\exp\left(-\frac{\delta k}{2\sigma^2}\right) & \text{if } \delta > \frac{2\sigma^2}{k} \end{cases}.$$

Let us choose $\delta = \frac{4\sigma^2 \ln(T|\mathcal{A}(\rho)|)}{k}$, such that $\delta > \frac{2\sigma^2}{k}$. Therefore, with probability greater than $1 - 2T^{-2}|\mathcal{A}(\rho)|^{-2}$, (49) $\leq \frac{4\sigma^2 \ln(T|\mathcal{A}(\rho)|)}{k}$.

$\square$

# E  Final Regret Calculation

*Proof of Theorem 6.1.* When we sum over $\rho \in \mathcal{P}$, we mean to refer to all balls over trial that are ever active, i.e. a member of $\mathcal{P}$ at any point of the algorithm within the time horizon $T$.

By using a similar argument to Lemma B.6, the regret from initial clustering is bounded above by

$$\frac{304 \cdot 5431 \cdot \sigma^2 K \ln(TK)}{L^2} = O\left(\frac{\sigma^2 K \ln(TK)}{L^2}\right),$$

where we used the property that regret is bounded above by 1 in every trial step because the expected reward function $f$ outputs values in $[0, 1]$.

Next we bound the expected regret after the initial clustering conditioned on the good events $\mathcal{G}$. We split the regret into two terms,

$$\mathbb{E}\left[\text{regret after initial clustering} \mid \mathcal{G}\right] \tag{56}$$

$$= \mathbb{E}\left[\left.\sum_{t=1}^{T}\sum_{\rho\in\mathcal{P}}\mathbb{I}\left(\rho_t = \rho\right)\left(f^*(x_t) - f_{a_t}(x_t)\right)\ \right|\ \mathcal{G}\right] \tag{57}$$

$$= \mathbb{E}\left[\left.\sum_{t=1}^{T}\sum_{i=1}^{i_{max}-1}\sum_{\rho\in\mathcal{P}}\mathbb{I}\left(\rho_t = \rho, \Delta(\rho) = 2^{-i}\right)\left(f^*(x_t) - f_{a_t}(x_t)\right)\ \right|\ \mathcal{G}\right] \tag{58}$$

$$+ \mathbb{E}\left[\left.\sum_{t=1}^{T}\sum_{i=i_{\max}}^{\infty}\sum_{\rho\in\mathcal{P}}\mathbb{I}\left(\rho_t = \rho, \Delta(\rho) = 2^{-i}\right)\left(f^*(x_t) - f_{a_t}(x_t)\right)\ \right|\ \mathcal{G}\right] \tag{59}$$

By Lemma B.5, (59) is bounded above by

$$\mathbb{E}\left[\left. 20L2^{-i_{\max}}\sum_{t=1}^{T}\sum_{i=i_{\max}}^{\infty}\sum_{\rho\in\mathcal{P}}\mathbb{I}\left(\rho_t = \rho, \Delta(\rho) = 2^{-i}\right)\ \right|\ \mathcal{G}\right] \leq 20L2^{-i_{\max}}T$$

Due to the fact that our algorithm always halved the context width, any ball $\rho$ with $\Delta(\rho) = 2^{-i}$ must have context width $w_i(\ell) = [(\ell-1)2^{-i}, \ell 2^{-i}]$ for some $\ell \in [2^i]$. As a result, (58) equals

$$\mathbb{E}\left[\left.\sum_{i=1}^{i_{max}-1}\sum_{\ell=1}^{2^i}\sum_{\rho\in\mathcal{P}}\mathbb{I}\left([c_0(\rho), c_1(\rho)] = w_i(\ell)\right)\sum_{t=1}^{T}\mathbb{I}\left(\rho_t = \rho\right)\left(f^*(x_t) - f_{a_t}(x_t)\right)\ \right|\ \mathcal{G}\right]$$

By Lemma B.5, conditioned on the good events $\mathcal{G}$, for any $\rho \in \mathcal{P}$ such that $\Delta(\rho) \leq \frac{1}{4}$,

$$\max_{(x,a)\in\rho}\left(f^*(x) - f_a(x)\right) \leq 20L\Delta(\rho).$$

This implies that for a context width $c_0(\rho), c_1(\rho)$, we can eliminate all arms $a \in [K]$ that are "extremely suboptimal", satisfying

$$\min_{x\in[c_0(\rho), c_1(\rho)]}\left(f^*(x) - f_a(x)\right) > 20L\Delta(\rho).$$

Let $\kappa(x)$ denote the suboptimality gap at context $x$, i.e. the difference between the optimal set of arms and the next optimal set of arms,

$$\kappa(x) = f^*(x) - \sup_{a\in[K]} f_a(x)\mathbb{I}\left(f_a(x) \neq f^*(x)\right).$$

If $\kappa(x) > 20L2^{-i}$ for all $x \in w_i(\ell)$, Lemma B.5 implies that conditioned on the good event $\mathcal{G}$, the regret incurred by any ball $\rho$ for which $[c_0(\rho), c_1(\rho)] \subseteq w_i(\ell)$ must be zero, as it must contain only optimal arms.

Thus we can reduce (58) to

$$\sum_{i=1}^{i_{max}-1}\sum_{\ell=1}^{2^i}\mathbb{I}\left(\min_{x\in w_i(\ell)}\kappa(x) \leq 20L2^{-i}\right)\mathbb{E}\left[\left.\sum_{\rho\in\mathcal{P}}\mathbb{I}\left([c_0(\rho), c_1(\rho)] = w_i(\ell)\right)\sum_{t=1}^{T}\mathbb{I}\left(\rho_t = \rho\right)\left(f^*(x_t) - f_{a_t}(x_t)\right)\ \right|\ \mathcal{G}\right]$$

By Lemmas B.5, B.4, and B.6, and using a trivial upper bound that $|\mathcal{A}(\rho)| \leq K$,

$$\mathbb{E}\left[\sum_{\rho \in \mathcal{P}} \mathbb{I}\left([c_0(\rho), c_1(\rho)] = w_i(\ell)\right) \left(\sum_{t=1}^{\tau_f(\rho)} \mathbb{I}\left(\rho_t = \rho\right) \left(f^*(x_t) - f_{a_t}(x_t)\right) + \sum_{t=\tau_f(\rho)+1}^{T} \mathbb{I}\left(\rho_t = \rho\right) \left(f^*(x_t) - f_{a_t}(x_t)\right)\right) \Bigg| \mathcal{G}\right]$$

$$\leq \mathbb{E}\left[\sum_{\rho \in \mathcal{P}} \mathbb{I}\left([c_0(\rho), c_1(\rho)] = w_i(\ell)\right) \left(20L\Delta(\rho)\left(\frac{6\sigma^2 \ln(T)}{L^2\Delta^2(\rho)} + 2\right) + 10L\Delta(\rho)\left(\frac{304 \cdot 5431 \cdot \sigma^2 |\mathcal{A}(\rho)| \ln(T|\mathcal{A}(\rho)|)}{L^2\Delta^2(\rho)}\right)\right) \Bigg| \mathcal{G}\right]$$

$$= O\left(\frac{\sigma^2 \ln(TK)}{L2^{-i}} \mathbb{E}\left[\sum_{\rho \in \mathcal{P}} \mathbb{I}\left([c_0(\rho), c_1(\rho)] = w_i(\ell)\right) |\mathcal{A}(\rho)| \Bigg| \mathcal{G}\right]\right)$$

The algorithm guaranatees that the active balls always form a partition of the context-arm space, and the balls are strictly nested in a hierarchy; as a result, the arms in different balls $\rho \in \mathcal{P}$ constrained to the same context width must be disjoint.

By Lipschitzness,

$$\min_{x \in w_i(\ell)} \left(f^*(x) - f_a(x)\right) \leq 20L2^{-i} \implies \max_{x \in w_i(\ell)} \left(f^*(x) - f_a(x)\right) \leq 22L2^{-i},$$

such that

$$\mathbb{I}\left(\min_{x \in w_i(\ell)} \left(f^*(x) - f_a(x)\right) \leq 20L2^{-i}\right) \leq \mathbb{I}\left(\left(f^*(2^{-i}\ell) - f_a(2^{-i}\ell)\right) \leq 22L2^{-i}\right).$$

For $i \geq 2$, by Lemma B.5, conditioned on the good events $\mathcal{G}$,

$$\mathbb{E}\left[\sum_{\rho \in \mathcal{P}} \mathbb{I}\left([c_0(\rho), c_1(\rho)] = w_i(\ell)\right) |\mathcal{A}(\rho)| \Bigg| \mathcal{G}\right] \leq \sum_{a \in [K]} \mathbb{I}\left(\min_{x \in w_i(\ell)} \left(f^*(x) - f_a(x)\right) \leq 20L2^{-i}\right)$$

$$\leq \sum_{a \in [K]} \mathbb{I}\left(\left(f^*(2^{-i}\ell) - f_a(2^{-i}\ell)\right) \leq 22L2^{-i}\right).$$

which upper bounds the number of arms that are ever subpartitioned into a ball of context width $w_i(\ell)$.

Let us denote

$$M_i = \sum_{\ell=1}^{2^i} \mathbb{I}\left(\min_{x \in w_i(\ell)} \kappa(x) \leq 20L2^{-i}\right) \sum_{a \in [K]} \mathbb{I}\left(\left(f^*(2^{-i}\ell) - f_a(2^{-i}\ell)\right) \leq 22L2^{-i}\right).$$

The scaling of this quantity depends on the local geometry amongst the arms with respect to the expected reward functions.

To get the final regret bound, we use Lemmas B.1 and B.1 to bound the probability that the good event $\mathcal{G}$ is violated,

$$\mathbb{E}\left[R(T)\right] \tag{60}$$

$$\leq \mathbb{E}\left[\text{regret from initial clustering}\right] + T\mathbb{P}(\neg\mathcal{G}) + \mathbb{E}\left[\text{regret after initial clustering} \mid \mathcal{G}\right] \tag{61}$$

$$\leq O\left(\frac{\sigma^2 K \ln(TK)}{L^2}\right) + T\left(2T^{-1} + 4T^{-1}\right) + 20LT2^{-i_{\max}} \tag{62}$$

$$+ O\left(\sum_{i=1}^{i_{max}-1} \frac{\sigma^2 M_i \ln(TK)}{L2^{-i}}\right) \tag{63}$$

$$= O\left(\frac{\sigma^2 K \ln(TK)}{L^2} + \min_{i_{\max} \in \mathbb{Z}_+} \left(LT2^{-i_{\max}} + \sum_{i=1}^{i_{max}-1} \frac{\sigma^2 M_i \ln(TK)}{L2^{-i}}\right)\right) \tag{64}$$

$$\square$$

**Finite Types**  Suppose that the reward functions for the $K$ arms, $\{f_a\}_{a \in [K]}$ only takes $M$ different values. Essentially, this implies that there are $\Theta$ different types of arms, but we don't know the arm types a priori. Within each type, the reward function is exactly the same. Let us denote the type of arm $a$ with $\theta_a \in [\Theta]$, and we define function $g : [0,1] \times [\Theta] \to [0,1]$ such that $f_a(x) = g(x, \theta)$. Let us define
$$\mu_\kappa(z) := \mu(\{x \in [0,1] \ s.t. \ \kappa(x) \le z\})$$
where $\mu$ is the Lebesgue measure.

Naively, $\sum_{a \in [K]} \mathbb{I}\left((f^*(2^{-i}\ell) - f_a(2^{-i}\ell)) \le 22L2^{-i}\right) \le K$. Furthermore, it will minimally be equal to the number of arms of the optimal type. If there are roughly proportional number of arms in each type, then this expression is also lower bounded by order $K$, thus it is sufficient to upper bound the expression by $K$. Then $M_i$ can be upper bounded by
$$M_i = K \sum_{\ell=1}^{2^i} \mathbb{I}\left(\min_{x \in w_i(\ell)} \kappa(x) \le 20L2^{-i}\right) \le \frac{K\mu_\kappa(22L2^{-i})}{2^{-i}},$$
which follows from the fact that $\kappa(x)$ must be a $2L$-Lipschitz function.

Let us denote the type of arm $a$ with $\theta_a \in [\Theta]$, and we define function $g : [0,1] \times [\Theta] \to [0,1]$ such that $f_a(x) = g(x, \theta)$. Let $\theta^*(x)$ denote the set of arm types that are optimal at context $x$,
$$\theta^*(x) = \{\theta \in [\Theta] \ s.t. \ g(x, \theta) = f^*(x)\}$$

In the finite types setting, the optimal policy corresponds to partitioning the context space $[0,1]$ into a set of intervals, $\mathcal{S}^*$, such that across each interval $\int \in \mathcal{S}^*$, the optimal policy does not change, i.e. $\theta^*(x) = \theta^*(x')$ for all $(x, x') \in \int \times \int$. Note that at each endpoint of $\int$, it must be that $\kappa(x) = 0$, as the fact that the policy changes and the reward functions are Lipschitz will imply that either an optimal arm becomes suboptimal, or a suboptimal arm becomes optimal, but the change must happen "smoothly" due to Lipschitzness. This also implies that for some $\gamma$ arbitrarily close to 0, if $x$ is an endpoint of any interval, either $\kappa(x + \gamma) > 0$ or $\kappa(x - \gamma) > 0$.

Let us assume that $\kappa(x)$ decreases linearly fast nearby the points where the optimal policy changes, so that for some constant $L'$,
$$\mu_\kappa(22L2^{-i}) \le \frac{22L2^{-i}}{L'} \times |\mathcal{S}^*|.$$
Then it follows by plugging into the main theorem that the regret is upper bounded by
$$\mathbb{E}\left[R(T)\right] = O\left(\frac{\sigma^2 K \ln(TK)}{L^2} + \min_{i_{\max} \in \mathbb{Z}_+}\left(LT2^{-i_{\max}} + \sum_{i=1}^{i_{\max}-1} \frac{22\sigma^2 |\mathcal{S}^*| K \ln(TK)}{L'2^{-i}}\right)\right).$$
(65)

By choosing
$$i_{\max} = \frac{1}{2}\log\left(\frac{L'LT}{22\sigma^2 |\mathcal{S}^*| K \ln(TK)}\right),$$
it follows that
$$\mathbb{E}\left[R(T)\right] \le O\left(\frac{\sigma^2 K \ln(TK)}{L^2} + \sqrt{\frac{\sigma^2 |\mathcal{S}^*| LTK \ln(TK)}{L'}}\right)$$
(66)

**Lipschitz arm geometry**  Suppose that each arm $a$ is associated to a latent variable $\theta_a \in [0,1]$, and the expected reward function $f_a(x) = g(x, \theta_a)$, where $g : [0,1] \times [0,1] \to [0,1]$ is a $L$-Lipschitz function with respect to both the contexts and the arm latent variables,
$$g(x, \theta) - g(x', \theta') \le L(|x - x'| + |\theta - \theta'|).$$

If we assume that the arm latent variables are uniformly spread out, $\{\theta_a\} = \{\frac{i}{K}\}_{i \in [K]}$, then
$$M_i = \sum_{\ell=1}^{2^i} \mathbb{I}\left(\min_{x \in w_i(\ell)} \kappa(x) \le 20L2^{-i}\right) \sum_{a \in [K]} \mathbb{I}\left((f^*(2^{-i}\ell) - g(2^{-i}\ell, \theta_a)) \le 22L2^{-i}\right) \quad (67)$$
$$\le \sum_{j \in [K]} \sum_{\ell=1}^{2^i} \mathbb{I}\left((f^*(2^{-i}\ell) - g(2^{-i}\ell, \frac{j}{K})) \le 22L2^{-i}\right), \quad (68)$$

which is a discrete approximation to the area of the arm-context space for which the suboptimality gap is at most $22L2^{-i}$. We can visualize $\sum_{\ell=1}^{2^i} M_i(\ell)$ by considering the contour plot of $f^*(x) - g(x,\theta)$, and counting how many grid points $\{(2^{-i}\ell, \frac{j}{K})\}_{\ell \in [2^i], j \in [K]}$ are lower than $22L2^{-i}$. For large $i$ and $K$, this is approximately equal to $2^i K \mu(\{(x,\theta) : g(x,\theta) - f^*(x) \geq -22L2^{-i}\})$, where $\mu$ is the Lebesgue measure. The curve at the lowest level of the contour plot corresponds to the set $\{(x,\theta) \, s.t. \, g(x,\theta) - f^*(x) = 0\}$, which contains for each context $x$ the set of arm latent variables $\theta$ that optimize the expected reward function. The final regret bound thus depends on the local measure/smoothness of the joint reward function.

To give a concrete example, we compute a bound for the reward function used in the simulation, where $g(x,\theta) = 1 - L|x - \theta|$ for some $L \in (0,1)$.

$$
\begin{aligned}
M_i &\leq \sum_{\ell=1}^{2^i} \sum_{a \in [K]} \mathbb{I}\left((f^*(2^{-i}\ell) - f_a(2^{-i}\ell)) \leq 22L2^{-i}\right) \\
&= \sum_{\ell=1}^{2^i} \sum_{j \in [K]} \mathbb{I}\left((f^*(2^{-i}\ell) - g(2^{-i}\ell, \frac{j}{K})) \leq 22L2^{-i}\right) \\
&\leq \sum_{\ell=1}^{2^i} \sum_{j \in [K]} \mathbb{I}\left(L|2^{-i}\ell - \frac{j}{K}| \leq 22L2^{-i}\right) \\
&\leq \sum_{\ell=1}^{2^i} \sum_{j \in [K]} \mathbb{I}\left(j \in [K(2^{-i}\ell - 22 \cdot 2^{-i}), K(2^{-i}\ell + 22 \cdot 2^{-i})]\right) \\
&\leq \sum_{\ell=1}^{2^i} 44 \cdot 2^{-i} K \\
&\leq 44K
\end{aligned}
$$

By plugging this into the main theorem, it follows that

$$
\mathbb{E}\left[R(T)\right] \quad = O\left(\frac{\sigma^2 K \ln(TK)}{L^2} + \min_{i_{\max} \in \mathbb{Z}_+}\left(LT2^{-i_{\max}} + \sum_{i=1}^{i_{max}-1} \frac{\sigma^2 K \ln(TK)}{L2^{-i}}\right)\right). \quad (69)
$$

Choosing

$$
i_{\max} = \frac{1}{2} \log\left(\frac{20L^2 T}{\sigma^2 K \ln(TK)}\right),
$$

results in

$$
\mathbb{E}\left[R(T)\right] \leq O\left(\frac{\sigma^2 K \ln(TK)}{L^2} + \sqrt{\sigma^2 KT \ln(TK)}\right). \quad (70)
$$

## F   Additional Simulation Results and Discussion

We test our algorithm on a model with 50, 100, 200 arms and a context space of $[0,1]$. Each arm $a$ corresponds to a parameter $\theta_a$ uniformly spaced out within $[0,1]$. The expected reward for arm $a$ and context $x$ is

$$
f_a(x) := g(x,\theta_a) = 1 - \left|x - 4\min_{z \in \{0,0.5,1\}} |\theta_a - z|\right|.
$$

This function is periodic with respect to $\theta$, and can be depicted as a zigzag. Our distance estimate $\hat{\mathcal{D}}_u^v(a,a')$ approximates $\mathcal{D}_u^v(a,a')$, which is defined with respect $f_a$ and $f_{a'}$ directly and does not depend on $\theta_a$. Consider a measure preserving transformation that maps $\theta_a$ to $\phi_a = 4\min_{z \in \{0,0.5,1\}} |\theta_a - z|$, such that the reward function is equivalently described by $f_a(x) = 1 - |x - \phi_a|$. An algorithm which partitions with respect to $\mathcal{D}_u^v(a,a')$ would be agnostic to such a transformation, as opposed to an algorithm which depends on a metric defined with respect to the arm's representation, which would perform worse on $\theta_a$ than $\phi_a$.

In a sequence of $T$ trials we uniformly randomly sample from the context space and reveal it to the algorithm. The algorithm selects what it considers to be the best arm in each trial based on the context revealed. Then simulation reveals a noisy payoff (i.e., $f_a(x) + \sigma$) to the algorithm. The task in the simulation setup[1] is for our algorithm to learn the optimal arm for different contexts.

We benchmark the performance of our Approx-Zooming algorithm against three variations:

- *Approx-Zooming -With-True-Reward-Function*: We give the Approx-Zooming algorithm oracle access to evaluate $\mathcal{D}_u^v(a, a')$ at no cost, which is used to subpartition whenever a ball is flagged.
- *Approx-Zooming -With-Similarity-Metric*: We give the Approx-Zooming algorithm oracle access to evaluate $|\theta_a - \theta_{a'}|$ at no cost, which is used to subpartition whenever a ball is flagged.
- *Approx-Zooming -With-No-Arm-Similarity*: This naive variant uses no arm similarities, estimating each arm's reward independently. The context space is adaptively partitioned via our algorithm.

The possible combinations of the parameters we have chosen are as follows. For all algorithms we have evaluated for $[50, 100, 200]$ arms over $[10,000, 50,000, 100,000]$ trials for $\sigma$ values $[1e - 1, 1e - 2, 1e - 3]$. We chose the model parameters that led to the highest average cumulative reward in each baseline algorithm based on a selected set of parameter permutations. There were no hyper parameters to be considered for Approx-Zooming-No-Arm-Similarity. As per the implementation of the algorithms Approx-Zooming-Similarity-Metric and Approx-Zooming-True we had to specify a starting bias (from $0.25, 0.5, 1$), starting distance threshold (from $0.25, 0.5, 1, 10$) within a partition and the context discretization mapping (from $5, 6, 7$). In addition to the above parameters as per the implementation of Approx-Zooming algorithm we had to specify the number of neighbours. The final $k$ nearest neighbours for the Approx-Zooming algorithm was a function of a minimum value we specified (from $3, 4, 5$), a global sampling constant (from $1, 2, 3$), the number of neighbours we specified explicitly (from $5, 10, 15$) and the context width. We evaluated the algorithms for some of these permutations and selected the best hyper-parameter configuration [2]. For all algorithms the flagging rule is set to $n_t(\rho) \geq 4\ln(T)/\Delta^2$, and for results reported in this paper $\sigma$ is set to either $1e - 3$ or $1e - 2$. For Approx-Zooming , $k$ was set to 10. We set the number of trials $T$ to $100,000$ as all the algorithms had converged to their optimal point by then. We present results for the three simulation settings:(1) 50 arms with noise $\sigma = 1e - 3$, (2) 100 arms with $\sigma = 1e - 3$ and (3) 200 arms with $\sigma = 1e - 2$.

In figure 3, we plot the average cumulative reward over the trials, i.e. $\frac{1}{T}\sum_{t=1}^{T} \pi_t$, where $T$ is the total number of trials and $\pi_t \in (0, 1)$ is the reward observed in the $t^{\text{th}}$ trial for different simulation settings. As we can see, the oracle variant of the algorithm that uses the true reward function to calculate $\mathcal{D}_u^v(a, a')$ performs the best on all three plots. Our Approx-Zooming algorithm has a heavy cost up front due to the clustering of the arms globally, but the algorithm improves over the time horizon as it learns the correct arm similarities. The oracle variant which uses the similarity metric $|\theta_a - \theta_{a'}|$ performs worse than the true $\mathcal{D}_u^v(a, a')$ variant, as it does not account for the periodic nature of the function.

In figure 4, figure 5 and figure 6 we plot the frequencies an arm is selected in different contexts over the $T$ trials in our three simulation settings. Each of the four plots correspond to averaging the frequency over $T/4$ trials across the time horizon. The x-axis refers to the context space, and the y-axis refers to the set of arms. As we see, the algorithms do generally learn to play the optimal policy, which corresponds to the zigzag shape. We can verify that in our algorithm initially the frequency plot is very blurry, indicating that it is spreading out the samples over time. As time progresses our algorithm indeed learns the similarities, which is depicted by the shape sharpening. The Approx-Zooming-True algorithm is given the true distance function, and we can see that indeed it is the sharpest curve. Approx-Zooming-Similarity-Metric, which is using the metric representation narrows in slower than the algorithm given true distance function. Approx-Zooming-No-Arm-Similarity which learns each arm separately initially finds a small set of arms that plays across the context and as a result takes more time to converge to the optimal policy.

The algorithm which learns each arm separately takes more time to converge to the optimal policy compared to all the other methods. Therefore, we can conclude that given a large arm set it is

(a) Simulation setup : 50 arms and $\sigma = 1e - 3$

(b) Simulation setup : 100 arms and $\sigma = 1e - 3$

(c) Simulation setup : 200 arms and $\sigma = 1e - 2$

Figure 3: Avg. cumulative reward vs. number of trials

important to use similarities amongst arms to find the optimal arm. Furthermore, we observed the algorithm using the metric representation narrows in slower than the algorithm given true distance function. Therefore, we argue if a metric space is used to find similarities in the arm spaces it needs to be carefully chosen to represent the reward distribution, which is not a trivial task. In contrast, our approach relies on samples from the reward distribution and learns the latent structure avoiding the difficulty of choosing a suitable metric. However, our approach needs to carefully tune the parameter $k$ to avoid unnecessary sampling for similarity estimation. We anticipate that the benefits of learning the metric only dominates in regimes where the number of arms is large and the time horizon is sufficiently long. Below we include similar plots for other parameters of the problem, in particular we analysed how Approx-Zooming performs compared to other benchmark algorithms when we have smaller number of arms but smaller number of trials with higher values for $\sigma$.

In table 1 we plot the final cumulative reward for each of the benchmark algorithms after 10000, 50000, 100000 trials. We have measured the cumulative reward for 50, 100, 200 arms and where $\sigma$ was set to either $1e - 1, 1e - 2$, or $1e - 3$. The hyper parameter values used in each of the benchmark algorithms for this experiment are as follows. For Approx-Zooming algorithm the starting bias was set to $0.25$ and the starting distance threshold was set to $0.5$. The number of neighbours were set to $10$. For the Approx-Zooming-Similarity-Metric algorithm the starting bias was set to $0.25$ and the starting distance threshold was set to $10$. For the Approx-Zooming-True algorithm the starting bias was set to $0.25$ and the starting distance threshold was set to $0.07$. As can be seen in table 1 for 50 arms or 100 arms we see a lower cumulative reward for Approx-Zooming compared to other benchmark algorithms when $\sigma = 1e - 1$ or $1e - 2$ and the total number of trials is $10,000$, which suggests that the cost due to the added extra exploration may exceed the gain from learning the metric when $\sigma$ is large or the total number of trials is small. This observation is aligned with the click through rates we report in figure 3 and illustrates that the benefits of learning the metric only dominates in regimes where the number of arms is large and the time horizon is sufficiently long.

(a) Approx Zooming

(b) Approx-Zooming-True

(c) Approx-Zooming-Similarity-Metric

(d) Approx-Zooming-No-Arm-Similarity

Figure 4: Arm Frequency Plots For 50 Arms

(a) Approx Zooming

(b) Approx-Zooming-True

(c) Approx-Zooming-Similarity-Metric

(d) Approx-Zooming-No-Arm-Similarity

Figure 5: Arm Frequency Plots For 100 Arms

(a) Approx Zooming

(b) Approx-Zooming-True

(c) Approx-Zooming-Similarity-Metric

(d) Approx-Zooming-No-Arm-Similarity

Figure 6: Arm Frequency Plots For 200 Arms

| Number of arms | Total number of trials | $\sigma$ | Approx-Zooming | Approx-Zooming-True | Approx-Zooming-Similarity-Metric | Approx-Zooming-No-Arm-Similarity |
|---|---|---|---|---|---|---|
| 50 | | | 0.789319 | 0.865815 | 0.828892 | 0.811413 |
| 100 | 10000 | 0.1 | 0.794965 | 0.836930 | 0.810921 | 0.776608 |
| 200 | | | 0.663524 | 0.838799 | 0.788084 | 0.757901 |
| 50 | | | 0.905667 | 0.911928 | 0.900178 | 0.861530 |
| 100 | 50000 | 0.1 | 0.895329 | 0.897861 | 0.882766 | 0.838076 |
| 200 | | | 0.850035 | 0.905815 | 0.848488 | 0.815208 |
| 50 | | | 0.926956 | 0.931452 | 0.914195 | 0.889772 |
| 100 | 100000 | 0.1 | 0.917762 | 0.911701 | 0.906180 | 0.859232 |
| 200 | | | 0.909856 | 0.918141 | 0.881553 | 0.836007 |
| 50 | | | 0.816736 | 0.862454 | 0.828892 | 0.811956 |
| 100 | 10000 | 0.01 | 0.762866 | 0.833096 | 0.810921 | 0.777493 |
| 200 | | | 0.735657 | 0.838271 | 0.788084 | 0.760152 |
| 50 | | | 0.893435 | 0.911078 | 0.903131 | 0.861970 |
| 100 | 50000 | 0.01 | 0.883395 | 0.897688 | 0.882611 | 0.837934 |
| 200 | | | 0.840253 | 0.908507 | 0.850934 | 0.814603 |
| 50 | | | 0.926705 | 0.931727 | 0.913347 | 0.891015 |
| 100 | 100000 | 0.01 | 0.912631 | 0.911363 | 0.906772 | 0.858644 |
| 200 | | | 0.907565 | 0.918062 | 0.881263 | 0.836181 |
| 50 | | | 0.843526 | 0.863722 | 0.837772 | 0.811940 |
| 100 | 10000 | 0.001 | 0.767987 | 0.833951 | 0.812152 | 0.776157 |
| 200 | | | 0.703802 | 0.837280 | 0.787980 | 0.760589 |
| 50 | | | 0.892600 | 0.910488 | 0.902087 | 0.864150 |
| 100 | 50000 | 0.001 | 0.884865 | 0.897461 | 0.882370 | 0.837778 |
| 200 | | | 0.866195 | 0.908066 | 0.850165 | 0.814999 |
| 50 | | | 0.927094 | 0.931463 | 0.913255 | 0.891716 |
| 100 | 100000 | 0.001 | 0.908725 | 0.911319 | 0.905410 | 0.859420 |
| 200 | | | 0.910485 | 0.916554 | 0.881567 | 0.836579 |

Table 1: The Cumulative Reward For Different Number Of Arms = $[50, 100, 200]$, Trials = $[10K, 50K, 100K]$ and $\sigma = [1e-1, 1e-2, 1e-3]$