[Reviews · NeurIPS 2019]

Reviewer 1



The paper is clear, and makes efforts to highlight the behavior of the proposed algorithm (value of the regret bound for some specific settings, experiments measuring the impact of the metric). The comparison to other settings may still be enforced. Typically, I would appreciate knowing the theoretical upper-bound of the regret of Approx-Zooming-With-No-Arm-Similarity. In the experimental part, I would also appreciate the comparison to include some state of the art algorithms. What would be the empirical results of a gaussian process-based bandit? It would also be interesting to have results on datasets used by other contextual/similarity-based bandits (except that these datasets use a context in R^d). Finally, it's surprising to have a context space of dimension 1. Extending the algorithm to R^d setting seems strait-forward. It would explode the time before sub-partitioning, but would it have more disadvantages? Would the R^d setting be more difficult to analyse? __________________________ # POST REBUTTAL I thank the author for their precise and clear rebuttal. It strengthens my positive opinion wrt. the paper. Regarding Gaussian process-based bandit, I was expecting a gaussian process per arm, but the version requiring an oracle access to a metric among the arms would also be informative wrt. to the efficiency of sub-partitioning. Regarding dimension-d context, given its necessity for most applications, the more the paper includes results wrt. that setting, the better. I understand experiments may not be conducted in the limited timeline, and the regret bound in the rebuttal gives a preliminary answer to the impact of dimension. The next step could be to also give the regret bound with dimension-d context for both concrete examples: finite types, and Lipschitz wrt. continuous arm metric space.

Reviewer 2



Originality: This work extends the contextual zooming algorithms to deal with the case of unknown similarity measure over the arm space. This is achieved by clustering the arms according to an an estimate of the integrated squared distance between two arms, based on a k-nearest neighbor estimators of the reward functions. This work is a nice combination of existing ideas. Quality: I think that paper is technically sound, although I did not verify all the proofs in the appendix. Clarity: The paper is well written and the key ideas are nicely presented. Significance: Adapting to unknown regularity parameters and hidden structures in a data driven manner is an important topic in nonparametric statistics, and this paper presents a method of adapting to unknown structure in the arm-space for nonparametric contextual bandits. I think that this is a well executed incremental contribution. Minor Comments: (i) Line 100: 'et al.' instead of 'et al' (ii) Line 104: has d_c been defined? (iii) Line 160: 'Nature could apply a measure preserving transform ...' -- Can you please illustrate this point with an example. ___________________ POST AUTHOR RESPONSE: I have read the authors' response as well as the other reviews. I thank the authors for clarifying about the generalization of their algorithm to higher dimensions. I understand that it may not be possible to modify the algorithm to make it adaptive to the H\"older smoothness by incorporating techniques of Qian&Zhang (2016) within the limited time. I will retain my previous score of 6 for this submission.

Reviewer 3



This paper modifies upon the contextual zooming algorithm (reference [22]) to learn a nonparametric contextual bandits. When the metric among the arms is unknown, a partitioning in the space of estimated reward function is used. Theoretical guarantees for the regret is provided. The result does seem to make sense. Although I must admit that I didn't check all the proofs. One fact that seems curious to me is that the overall regret bound can get better when the estimate for the Lipschitz constant of the reward function is looser. For example, in Eq. (9), the overall regret scales inversely with L^2. I wonder whether this artifact of the subpartitioning step in the contextual zooming algorithm can be easily removed?

[Author Response · NeurIPS 2019]

We greatly appreciate the feedback of the reviewers. We discuss the specific concerns of the reviewers below.

Reviewer 1:

• A theoretical upper-bound of the regret of Approx-Zooming-With-No-Arm-Similarity is stated in [7] as
$O(KT^{\frac{d+1}{d+2}})$, where $d$ is the dimension of the context space. Note that this regret bound is linear in the number
of arms. We will include this discussion into the paper.

• We will include empirical results of a gaussian process-based bandit in the final paper. As one needs to specify
the covariance/kernel matrix, we could either fit a Gaussian process to each arm separately when the metric is
unknown, or we can construct the covariance matrix given oracle access to a metric among the arms.

• Generalizing to higher dimension: Yes, indeed our algorithm and analysis does extend to the general $d$-
dimensional setting, and we will include that into the final paper. The only change required algorithmically is
in the subpartitioning/clustering step. Let us define $C_d(q)$ to be the number of balls of radius $r/q$ needed to
cover a ball of radius $r$, which scales exponentially in the dimension $d$, e.g. $q^d$. Since we are now estimating
the reward function $f$ over a $d$-dimensional context space, the number of sub-regions of the context space that
need to be clustered will be $C_d(2)$, and the number of samples needed to guarantee that the $k$-nearest neighbor
samples are within distance $\frac{1}{16}$ radius, will be equal to $\tilde{O}(kC_d(32))$. To compute $\hat{\mathcal{D}}$, we will instead have a
$d$-dimensional summation over the subset of the context space. Once $\hat{\mathcal{D}}$ is computed, then the clustering of
arms will have the same computational cost, i.e. linear in number of arms to be clustered. The analysis can be
modified to account for the $d$-dimensional setting, and the final regret bound will look like

$$O\Big(C_d(2)C_d(32)\sigma^2 L^{-2}K\ln(TK) + \min_{i_{\max}\in\mathbb{Z}_+}\big(LT2^{-i_{\max}} + \sum_{i=1}^{i_{max}-1}C_d(2)C_d(32)\sigma^2 L^{-1}M_i 2^i \ln(TK)\big)\Big),$$

where $M_i$ instead sums over an $\epsilon$-net of the context space for $\epsilon = 2^{-i}$, and thus we may expect $M_i$ to grow
exponentially in $i \times d$, although modified with respect to the distribution of the reward function and the finite
arms. The growth of $M_i$ will dominate the regret bound with respect to the dependence on the dimension $d$.

Reviewer 2:

• Line 160: We will include an example, in fact the setup we chose for the simulation illustrates this as it is a
periodic function that could be rearranged to be simply linear rather than the depicted zigzag.

• Adapting to the smoothness parameter of the reward function: Indeed our algorithm can be generalized to
Holder continuous reward functions. If the smoothness parameter is known, then modifying the algorithm is
straightforward. We will look into the techniques of Qian and Yang (2016) for adaptivity to the smoothness.

• Generalizing to higher dimension: see response to reviewer 1 above.

Reviewer 3:

• Regret bound scaling with Lipschitz constant: There are two terms in the final regret bound. The first term
comes from the cost of the initial clustering, and as you pointed out it scales inversely with $L^2$. However, the
first term scales as $\tilde{O}(K)$, logarithmic in $T$, whereas the second term has a polynomial dependence on $T$, e.g.
$\tilde{O}(\sqrt{KT})$ in the given examples. For large $T$ and $K = o(T)$, the second term will dominate, which does not
scale inversely with the Lipschitz constant. To give more inutition though, the inverse dependence on $L$ in the
first term (only logarithmic scaling wrt $T$) is due to the fact that the clusters are constructed to satisfy that the
bias in the reward due to different arms in the same cluster is controlled to be on the same order as the bias in
reward from different contexts in the same set. For small Lipschitz constant, the reward varies less across the
same size context width, and thus the algorithm requires that arm distances are measured more precisely to
guarantee that the bias of the new ball is bounded by $L$ times the context width. This initial clustering of the
algorithm could be modified to allow for looser clusters, where the bias due to different arms in a cluster is
larger than the context width. This would remove the inverse dependence on $L$ in the first term, but would add
more phases of subpartitioning that would contribute regret to the summation in the second term.

• The regret bound obtained in the contextual zooming algorithm when the metric among the arms is available is
$O(T^{\frac{d+1}{d+2}})$, where $d$ is the dimension of the joint context-arm space, i.e. $d = d_c + d_a$ where $d_c$ is the dimension
of the context space and $d_a$ is the dimension of the arm space. We will include this into the final paper.

We will also address the minor typos/comments in the revision as well. Thank you for your detailed feedback!

[Meta-Review · NeurIPS 2019]

This paper proposed a nonparametric approach to contextual bandits that can adapt to unknown simple structure between the arms. The reviewers found the paper to be novel in how it combined existing ideas to solve an interesting problem. Additionally, the reviewers found the paper to be clearly written and of potential significance for the problem they tacked. Some minor concerns were raised, however, the authors seem to have addressed them in their response. The authors should incorporate any suggested edits by the reviewers as well as the promised updates in the author response.